# Provably Learning Representations under Generalized Dependency Structure

## Abstract

The presence of noise that depends on the latent variable poses a significant identifiability challenge. Addressing this issue, the standard solution in the literature assumes that the observational data satisfy the conditional independence property given the latent variables. However, this assumption might not be valid in practice. This work relaxes this foundational constraint. Specifically, we consider a *generalized dependency structure* in which the observations may exhibit arbitrary dependencies conditional on the latents. To establish identifiability guarantees, we introduce a two-step theoretical framework. First, we formulate the problem as a factor analysis model use perturbation theory to establish the subspace identifiability of the latent variables. Second, assuming the structural sparsity on the mixing function, or sufficient variability constraint in the latent space, we establish component-wise identifiability of each individual latent factor. Using these identifiability results, we develop an unsupervised approach that reliably uncovers the latent representations. Experiments on synthetic and real data verify our theoretical claims.

## 1 Introduction

Identification of latent variables underpins the true data generating processes, thus inspiring an extensive line of works on downstream tasks, such as transfer learning (Kügelgen et al., 2021; Kong et al., 2022; Xie et al., 2023; Li et al., 2024) and visual reasoning (Chen et al., 2024a;b; Kong et al., 2024). When there exists structured noise depending on the latent variables, which we term as ***dependent noise***, theoretical guarantees of identifiability become significant challenging to establish. Prior work (Hu, 2008) partially addresses this issue but fundamentally assumes that observations are conditionally independent given the latents. Somce recent works Zheng et al. (2025); Fu et al. (2025); Li et al. (2025c) extends the conditional independence setup to time-series data. However, many real-world systems might violate such assumption. For example, in chest X-ray based disease diagnosis, the goal is to infer a patient's latent lung-cancer state from the pixel intensities observed in an image. The patient's inspiratory level, which may itself depend on disease condition, introduces noise to pixel intensities. Partitioning the image into anatomical regions reveals that regional intensities remain spatially correlated even with conditioning on disease status. This highlights the need for a flexible and robust framework to identify the latent variables under ***generalized dependency structure***, *which allows dependencies among observations conditioning on latent variables.*

The existence of dependent noise within the generalized dependency structure necessitates explicitly disentangling the latent variable from noise for identifiability. The existing literature has yet to fully address this challenge. Several works treats the noise terms as known auxiliaries (Lachapelle et al., 2024b; Liang et al., 2023; Lachapelle et al., 2023; Liang et al., 2023; Lippe et al., 2023; Zheng et al., 2022; Yao et al., 2024; Lachapelle et al., 2024a; Li et al., 2025a; Song et al., 2024; Rajendran et al., 2024; Xu et al., 2024; Brady et al., 2025). One might argue that noise term can simply be absorbed into an expanded latent space. The approaches of (Kügelgen et al., 2021; Kong et al., 2022; Xie et al., 2023; Li et al., 2025b; Ng et al., 2025) pursue this route but require partitioning them into invariant / variant components across environments. Rather than partitioning the latent variables, a few methods rely on carefully designed structure between latent variable and the noise term. Specifically, Sun et al. (2025) assumes independence between noise and latent variables, while (Kong et al., 2023) requires that the relations between a pair of latent variables have to be sufficiently distinct. In this paper, we use the term *generalized dependency structure* to refer to data-generating

processes in which the observed variables are allowed to have arbitrary dependence conditioning on the latent variable, without the conditional independence restrictions imposed in Li et al. (2025c); Fu et al. (2025).

In contrast to previous methods, we present identifiability theory that uncovers the latent variables under generalized dependency structure, without requiring pre-specified auxiliary variables, latent variable partitions and carefully designed structure. Our analysis starts with establishing subspace identifiability, which separates the noise and latent variables in Theorem 1 through a spectral decomposition tailored to bounded perturbations. Building upon this foundational result, we introduce structural sparsity assumption in Theorem 1, which rigorously guarantees the component-wise identifiability of latent variables. Moreover, we present an alternative result through Theorem 2, which leverages sufficient variability across multiple domains to ensure component-wise identifiability. To the best of our knowledge, combining subspace identifiability (Theorem 1) with either structural sparsity (Theorem 1) or sufficient-variability assumptions (Theorem 2) yields the first general frameworks for reliably identifying latent variables under generalized dependency structures.

Leveraging these theoretical insights, we propose an unsupervised method, which utilizes a variational inference-based learning objective specifically designed to uncover latent variables. Our approach effectively models the intricate data-generating processes involving generalized dependency structures. Extensive experimental evaluations on both synthetic and real-world datasets showcase significant improvements over existing methods, thereby validating the robustness and effectiveness of our theoretical and methodological advancements.

## 2 PROBLEM SETTING

Let $\mathbf{x} \in \mathbb{R}^K$ denote the $K$ dimensional observation, $\mathbf{z} \in \mathbb{R}^N$ denote the latent variable, and $\epsilon \in \mathbb{R}^M$ denote the dependent noise. Also, we assume $p(\mathbf{z})$ is positive and smooth. Our data generating process is formulated by:

$$\mathbf{x} = g(\mathbf{z}, \epsilon), \quad \epsilon = e(\mathbf{z}, \eta) \tag{1}$$

We assume $g$ to be nonlinear, nonparametric, injective and smooth functions. $e$ denotes another nonlinear, nonparametric, injective and smooth function. $\eta$ denotes an independent exogenous variable sampled from $\mathcal{N}(0, 1)$.

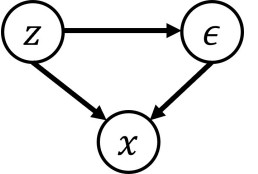

Figure 1: Visualization of the data generations of Eq. 1

Our primary goal is to identify latent variables $\mathbf{z}$ from observed data $\mathbf{x}$. To achieve this, we first introduce the definition of observational equivalence as follows:

**Definition 1** (**Observational Equivalence**). *Let the true data-generating process for the observed variables $\mathbf{x}$ be characterized by $\{g, e, p_{\mathbf{z}}(\mathbf{z}), p_{\epsilon}(\epsilon)\}$ as specified in Eq. 1. A learned model $\{\hat{g}, \hat{e}, p_{\hat{\mathbf{z}}}(\hat{\mathbf{z}}), p_{\hat{\epsilon}}(\hat{\epsilon})\}$ is said to be observationally equivalent to the true model if their induced data distributions match exactly, i.e.,*

$$p_{g,e,p_{\mathbf{z}},p_{\epsilon}}(\mathbf{x}) = p_{\hat{g},\hat{e},p_{\hat{\mathbf{z}}},p_{\hat{\epsilon}}}(\mathbf{x}) \tag{2}$$

Identifying $\mathbf{z}$ allows recovery of $g$ and $e$ up to certain indeterminacies, as we assume the injectivity of $g$ and $e$. Also, we assume there exists no latent confounders,

Suppose there exists an invertible and differentiable transformation $h$, we further define the notions of subspace identifiability and component-wise identifiability as follows:

**Definition 2.** *(Subspace Identifiability) The mapping $h$ is said to achieve subspace identifiability if there exists a permutation $\pi$ such that the following transformation holds: $\hat{\mathbf{z}} = h(\pi(\mathbf{z}))$, where $\hat{\mathbf{z}}$ denotes the estimation of $\mathbf{z}$.*

**Definition 3.** *(Component-wise Identifiability) For an individual component of the latent variable $\mathbf{z}^n$ ($\mathbf{n} \in [1, N]$), there exists a unique component $\hat{n}$ ($\hat{n} \in [1, N]$) of $\hat{\mathbf{z}}$ matches $\mathbf{z}^n$ up to a permutation $\pi$, such that $\hat{\mathbf{z}}^{\hat{n}} = h^{\hat{n}}(\mathbf{z}^{\pi(n)})$. Then $\mathbf{z}^n$ is component-wise identifiable.*

In this work, we breakdown the identifiability problem in two steps. First, we provide our findings on subspace identifiability in Section 4. Subsequently, we present the component-wise identifiability result in Section 5.

## 3 PRELIMINARIES

To situate our identifiability results within the existing literature, we first recapitulate and summarize the core assumptions from (Hu, 2008), which supports the identifiability of latent variables up to an invertible transformation. See Appendix B.1 for the detailed proof of the following.

Consider an alternative data generating process defined by: $\mathbf{x}' = g'(\mathbf{z}', \epsilon')$, $\epsilon' = e'(\mathbf{z}', \eta')$, where $\mathbf{z}'$ is a latent variable, $\epsilon'$ is an independent noise term, $\eta'$ is an auxiliary variable, $g'$ is an injective smooth function, and $e'$ is an injective and smooth function. $\mathbf{x}'$ can be decomposed into three disjoint parts $\mathbf{x}' = (\mathbf{x}'_A, \mathbf{x}'_B, \mathbf{x}'_C)$. Hu (2008) assumes *conditional independence* of $\mathbf{x}'$ given the latent variable $\mathbf{z}'$ by $p(\mathbf{x}'|\mathbf{z}') = p(\mathbf{x}'_A|\mathbf{z}')p(\mathbf{x}'_B|\mathbf{z}')p(\mathbf{x}'_C|\mathbf{z}')$. Under this characterization, $\mathbf{x}'$ can be interpreted as three conditionally independent measurements of $\mathbf{z}'$.

Identifying $\mathbf{z}'$ necessities disentangling $\epsilon$ from $\mathbf{z}$. Without further assumptions, this is impossible since $\epsilon'$ is inherently generated from $\mathbf{z}'$. To overcome this challenge, we leverage the spectral decomposition of the integral linear operator:

**Definition 4.** *Consider random variables $\mathbf{r}$ and $\mathbf{v}$ with supports $\mathcal{R}$ and $\mathcal{V}$ respectively. The linear integral operator $L_{\mathbf{r}|\mathbf{v}}$ maps a function $f \in F(\mathcal{V})$ to another function $L_{\mathbf{r}|\mathbf{v}} f \in F(\mathcal{R})$, defined by:* $(L_{\mathbf{r}|\mathbf{v}} f)(\mathbf{r}) = \int_{\mathcal{V}} p(\mathbf{r}|\mathbf{v}) f(\mathbf{v}) \, d\mathcal{V}, \quad \forall \mathbf{r} \in \mathcal{R}.$

where $p(\mathbf{r}|\mathbf{v})$ denotes the conditional density of $\mathbf{r}$ given $\mathbf{v}$. We consider $L_{\mathbf{r}|\mathbf{v}}$ to be well defined and bounded. With Definition 4 established, we impose the following assumptions:

**Assumption 1.** *The operators $L_{\mathbf{x}'_A|\mathbf{z}'}$ and $L_{\mathbf{x}'_A|\mathbf{x}'_B}$ are injective.*

**Assumption 2.** *for any $\underline{\mathbf{z}}' \neq \bar{\mathbf{z}}'$, the set $\mathbf{x}'_C : \{p(\mathbf{x}'_C|\underline{\mathbf{z}}') \neq p(\mathbf{x}'_C|\bar{\mathbf{z}}')\}$ has positive probability.*

Given Assumptions 1, Hu (2008) arrives at

$$L_{\mathbf{x}'_C; \mathbf{x}'_A | \mathbf{x}'_B} L_{\mathbf{x}'_A | \mathbf{x}'_B}^{-1} = L_{\mathbf{x}'_A | \mathbf{z}'} L_{\mathbf{x}'_C | \mathbf{z}'} L_{\mathbf{x}'_A | \mathbf{z}'}^{-1} \tag{3}$$

where the LHS involves only observable variables, and the RHS explicitly depends on the latent variable $\mathbf{z}'$. $L_{\mathbf{x}'_C|\mathbf{z}'}$ determines the eigenvalues of $L_{\mathbf{x}'_A|Z} L_{\mathbf{x}'_C|\mathbf{z}'} L_{\mathbf{x}'_A|\mathbf{z}'}^{-1}$, whose diagonal entries correspond to the conditional distributions $p(\mathbf{x}'_C|\mathbf{z}')$.

Eq. 3 suggests that each $\mathbf{z}'$ indexing a distinct conditional distribution of $p(\mathbf{x}'_C|\mathbf{z}')$. Under Assumption 2, where $p(\mathbf{x}'_C \mid \mathbf{z}')$ are distinct for different values of $\mathbf{z}'$, the eigenvalues are distinct. This allows a bijective mapping $h' : \mathcal{Z} \to \mathcal{Z}$ to permute $\mathbf{z}'$ while preserving the values of $p(\mathbf{x}'_C \mid \mathbf{z}')$. Therefore, the latent variable can only be recovered up to such a permutation, i.e., $\hat{\mathbf{z}}' = h'(\mathbf{z}')$, which yields the identifiability up to an invertible transformation $h'$.

This identifiability result foundamentally relies on the conditional independence $p(\mathbf{x}'|\mathbf{z}') = p(\mathbf{x}'_A|\mathbf{z}')p(\mathbf{x}'_B|\mathbf{z}')p(\mathbf{x}'_C|\mathbf{z}')$, which might be restrictive. Also, $h'$ might not meet Definition 2 since it does not have to be differentiable. In what follows, we address to identify $\mathbf{z}$ under the *generalized dependency structure* accommodating both conditional dependence across observations, as well as differentiable transformation between the estimated and true latent variables.

## 4 SUBSPACE IDENTIFIABILITY

Our work builds upon Definition 4, Assumptions 1 and 2. Let $\mathcal{X}$ denote the support of observed variables $\mathbf{x}$ of Eq. 1, which can be partitioned into three subsets $\{\mathbf{x_a}, \mathbf{x_b}, \mathbf{x_c}\}$. This work allows $\{\mathbf{x_a}, \mathbf{x_b}, \mathbf{x_c}\}$ remaining dependent conditioning on $\mathbf{z}$. Formally, this implies: $p(\mathbf{x}|\mathbf{z}) \neq p(\mathbf{x_a}|\mathbf{z})p(\mathbf{x_b}|\mathbf{z})p(\mathbf{x_c}|\mathbf{z})$, which consequently violates Eq. 3. As a result, we can only obtain $L_{\mathbf{x_a}, \mathbf{x_b}|\mathbf{x_c}} L_{\mathbf{x_a}|\mathbf{x_c}}^{-1} \neq L_{\mathbf{x_a}|\mathbf{z}} L_{\mathbf{x_b}|\mathbf{z}} L_{\mathbf{x_a}|\mathbf{z}}^{-1}$, where $L_{\mathbf{x_a}|\mathbf{x_c}}^{-1}$ and $L_{\mathbf{x_a}|\mathbf{z}}^{-1}$ exist by Assumption 1. A detailed explanation appears in the proof of Theorem 1 (Appendix B.2). To address this setting, we explicitly define a perturbation operator $Per$ to facility such inequality:

$$L_{\mathbf{x_a}|\mathbf{z}} L_{\mathbf{x_b}|\mathbf{z}} L_{\mathbf{x_a}|\mathbf{z}}^{-1} = L_{\mathbf{x_a}, \mathbf{x_b}|\mathbf{x_c}} L_{\mathbf{x_a}|\mathbf{x_c}}^{-1} + Per \tag{4}$$

where $Per \neq 0$ denotes deviations from $L_{\mathbf{x_a}, \mathbf{x_b}|\mathbf{x_c}} L_{\mathbf{x_a}|\mathbf{x_c}}^{-1}$. Our aim thus becomes to identify $\mathbf{z}$ from $L_{\mathbf{x_a}|\mathbf{z}} L_{\mathbf{x_b}|\mathbf{z}} L_{\mathbf{x_a}|\mathbf{z}}^{-1}$. Notably, the partition $\mathbf{x} = (x_a, x_b, x_c)$ is arbitrary; Our analysis only requires that there exist these partitions, and is invariant to any relabeling of these partitions.

With the problem formulation in Eq. 4 at hand, we present subspace identifiability result:

**Theorem 1.** *Consider observed variables $\mathbf{x} \in \mathbb{R}^K$ and the estimated latent variables $\hat{\mathbf{z}} \in \mathbb{R}^N$, suppose that there exist functions $\hat{g}$ and $\hat{e}$ satisfying the observational equivalence defined in Eq. 2, and the following assumptions hold:*

    *i For $\mathbf{x} = \{\mathbf{x_a}, \mathbf{x_b}, \mathbf{x_c}\}$, we allow the dependencies such that $p(\mathbf{x}|\mathbf{z}) \neq p(\mathbf{x_a}|\mathbf{z})p(\mathbf{x_b}|\mathbf{z})p(\mathbf{x_c}|\mathbf{z})$;*

    *ii The operators $L_{\mathbf{x_a}|\mathbf{z}}$, $L_{\mathbf{z}|\mathbf{x_c}}$, and $L_{\mathbf{x_a}|\mathbf{x_c}}$ are injective;*

    *iii The operator $L_{\mathbf{x_a},\mathbf{x_b}|\mathbf{x_c}}L_{\mathbf{x_a}|\mathbf{x_c}}^{-1}$ has distinct eigenvalues with cardinality equal to that of $L_{\mathbf{x_b}|\mathbf{z}}$;*

    *iv $L_{\mathbf{x_a}|\mathbf{z}}L_{\mathbf{x_b}|\mathbf{z}}L_{\mathbf{x_a}|\mathbf{z}}^{-1}$ is self-adjoint.*

    *v $\rho^i$ denotes the $i$-th eigenvalue of the operator $L_{\mathbf{x_a},\mathbf{x_b}|\mathbf{x_c}}L_{\mathbf{x_a}|\mathbf{x_c}}^{-1}$. Let $\kappa = \min_{i \neq j} \frac{|\rho^i - \rho^j| - \alpha}{2} \geq 0$ for some constant $\alpha > 0$, and $\overline{|Per|} < \kappa$, where $\overline{|Per|}$ denotes the upper bound of $Per$;*

    *vi Assumption $i \sim v$ results in the existence of an invertible transformation $\tilde{h} : \mathcal{Z} \to \mathcal{Z}$, such that $\tilde{\mathbf{z}} = \tilde{h}(\mathbf{z})$, where $\tilde{\mathbf{z}} \in \mathcal{Z}$. We further assume that $\exists M$ such that $M(L_{\mathbf{x_b}|\mathbf{z}}) = M(L_{\mathbf{x_b}|\tilde{h}(\mathbf{z})}) = t(\mathbf{z})$, where $t$ is a differentiable transformation.*

*then for $\tilde{h} \in \tilde{\mathcal{H}}$ and $t \in \mathcal{T}$ (where $\tilde{\mathcal{H}}$ and $\mathcal{T}$ are function classes, and $\tilde{\mathcal{H}} \cap \mathcal{T} \neq \emptyset$), if $h \in \tilde{\mathcal{H}} \cap \mathcal{T} \Rightarrow \hat{\mathbf{z}} = h(\mathbf{z}) = \tilde{h}(\mathbf{z}) = t(\mathbf{z})$. In other words, $\mathbf{z}$ must be subspace identified.*

**Proof sketch:** We detail the proof in the Appendix B.2, and this section summarizes the key steps. As previously described, Assumptions i and ii are used to derive Eq. 4. Furthermore, Assumption iii implies that, under certain conditions, $L_{\mathbf{x_b}|\mathbf{z}}$ can possess unique entries. To establish such uniqueness under perturbation, let $\rho^i$ and $\rho_\Lambda^i$ denote the $i$-th eigenvalues of $L_{\mathbf{x_a},\mathbf{x_b}|\mathbf{x_c}}L_{\mathbf{x_a}|\mathbf{x_c}}^{-1}$ and $L_{\mathbf{x_a}|\mathbf{z}}L_{\mathbf{x_b}|\mathbf{z}}L_{\mathbf{x_a}|\mathbf{z}}^{-1}$, respectively. Applying Weyl's inequality (Kato, 2013) to Eq. 4 under Assumption iv, we obtain $|\rho_\Lambda^i - \rho^i| \leq \overline{Per}$. Then, by Assumption v, if $\overline{|Per|} < \kappa$, all eigenvalues $\rho_\Lambda^i$ remain distinct for any $i \neq j$. Hence, $\overline{|Per|}$ quantifies the permissible perturbation tolerance under which $L_{\mathbf{x_a}|\mathbf{z}}L_{\mathbf{x_b}|\mathbf{z}}L_{\mathbf{x_a}|\mathbf{z}}^{-1}$ retains unique eigenvalues. The uniqueness of $L_{\mathbf{x_b}|\mathbf{z}}$ implies that permuting $\mathbf{z}$ would not affect the eigenvalues, i.e., there exists an invertible permutation $\tilde{h} : \mathcal{Z} \to \mathcal{Z}$ such that $\tilde{\mathbf{z}} = \tilde{h}(\mathbf{z})$. Finally, Assumption vi ensures there exists $h \in \tilde{\mathcal{H}} \cap \mathcal{T}$ that is both invertible and differentiable, satisfying the requirements of Definition 2.

**Remark:** Assumption i characterizes conditional dependencies among $\mathbf{x_a}, \mathbf{x_b}, \mathbf{x_c}$ given $\mathbf{z}$. Assumption ii is adopted by (Hu, 2008) as well. It ensures the existence of the inverse operators $L_{\mathbf{x_a}|\mathbf{z}}^{-1}$, $L_{\mathbf{z}|\mathbf{x_c}}^{-1}$ and $L_{\mathbf{x_a}|\mathbf{x_c}}^{-1}$ required in Eq. 4. To address the scaling and potential eigenvalue degeneracy of the operator $L_{\mathbf{x_a}|\mathbf{z}}L_{\mathbf{x_b}|\mathbf{z}}L_{\mathbf{x_a}|\mathbf{z}}^{-1}$, we introduce Assumption iii. Furthermore, Assumptions iv and v are crucial for controlling the perturbation term $Per$, thereby ensuring the distinctness of $L_{\mathbf{x_b}|\mathbf{z}}$. This distinctness is essential: if eigenvalues were to coincide, the spectral structure $L_{\mathbf{x_a}|\mathbf{z}}L_{\mathbf{x_b}|\mathbf{z}}L_{\mathbf{x_a}|\mathbf{z}}^{-1}$ would become ambiguous, and identifiability would be lost. The uniqueness of $L_{\mathbf{x_b}|\mathbf{z}}$ subsequently enables a permuting of $\mathbf{z}$ via a bijection $\tilde{h} : \mathcal{Z} \to \mathcal{Z}$. Notably, the bijection $\tilde{h}$ need not coincide with the transformation $h$ stipulated in Definition 2. Although the smoothness of $p(\mathbf{z})$ and the mixing function $g$ from Eq. 1 imply that $h$ is differentiable, establishing subspace identifiability requires that $h$ matches $\tilde{h}$ exactly. To formally guarantee this equivalence, we introduce Assumption vi.

## 5 COMPONENT-WISE IDENTIFIABILITY

Theorem 1 guarantees that $\hat{\mathbf{z}}$ is not a function of $\epsilon$, hence $\mathbf{z}$ and $\epsilon$ are disentangled. We now focus on identifying $\mathbf{z}$ in a component-wise manner. Let $J_g(\mathbf{z})$ denote the Jacobian of the mixing function $g$, and let $G \in \{0, 1\}^{K \times N}$ represent a binary adjacency matrix indicating connections from latent variables $\mathbf{z}$ to observed variables $\mathbf{x}$, where $G^{r,c} = 1$ suggests the existence of the relationship from $\mathbf{z}^c$ to $\mathbf{x}^r$. Hence, $G$ is interpreted as the binarized $J_g(\mathbf{z})$. We formally state our first main result regarding component-wise identifiability as follows:

**Corollary 1.** *Consider the true model $\{g, e, p(\mathbf{z}), p(\epsilon)\}$ and a learned model $\{\hat{g}, \hat{e}, p(\hat{\mathbf{z}}), p(\hat{\epsilon})\}$ that satisfy observational equivalence (Definition 1) and subspace identifiability (Theorem 1). Suppose the following assumptions and regularization conditions hold:*

A *Latent dimensions of $\mathbf{z}$ are independent: $p(\mathbf{z}) = \prod_{n=1}^{N} p(\mathbf{z}^n)$;*

B *For each dimension $n \in [1, N]$ of $\mathbf{z}$, there exist $\{\mathbf{z}^l\}_{l=1}^{|G^{n,:}|}$ such that:*

$$\text{span}\{J_g(\mathbf{z}^l)_{n,:}\}_{l=1}^{|G^{n,:}|} = \mathbb{R}_{G^{n,:}}^N, \quad \text{and} \quad [J_{\hat{g}}(\hat{\mathbf{z}}^l)_{n,:}]_{l=1}^{|\hat{G}^{n,:}|} \in \mathbb{R}_{\hat{G}^{n,:}}^N$$

C *For each $n \in [1, N]$, there exists a subset of indices $\mathcal{C}_k$ satisfying $\bigcap_{m \in \mathcal{C}_k} G^{m,:} = \{n\}$;*

D *Sparsity regularization: $|\hat{G}| \le |G|$*

*Then, $\hat{\mathbf{z}}$ must correspond component-wise to a permutation of the true latent variables $\mathbf{z}$.*

**Proof Sketch:** The complete proof is deferred to the Appendix B.3. Here we highlight key steps. First, observational equivalence and subspace identifiability (Theorem 1) imply $\hat{\mathbf{z}} = h(\mathbf{z})$, which leads to: $J_g(\mathbf{z}) = J_{\hat{g}}(\hat{\mathbf{z}}) J_h(\mathbf{z})$. By leveraging Assumption B along with the sparsity regularization condition D, we can prove that there exists a permutation between $\mathbf{z}$ and $\hat{\mathbf{z}}$. Subsequently, component-wise identifiability is proven by contradiction: any violation would contradict the structural sparsity assumption C.

**Remark:** Assumptions A is commonly employed in recent literature (Kong et al., 2022; Xie et al., 2023). Assumption B is introduced to ensure that the Jacobian spans the appropriate subspace. Previous works Zheng et al. (2022) leverages a similar assumption for their identifiability results under only the noise-free data generating process. In contrast, Theorem 1 forms the basis of Theorem 1, and their combination demonstrate the component-wise identifiability even under generalized dependency structure.

To relax Assumption A in Theorem 1, we can alternatively allow latent variables $\mathbf{z}$ to exhibit dependence via a known auxiliary domain variable $\mathbf{u}$. In other words, we assume conditional independence across dimensions of $\mathbf{z}$ given $\mathbf{u}$, i.e., $p(\mathbf{z}|\mathbf{u}) = \prod_{n=1}^{N} p(\mathbf{z}^n|\mathbf{u})$. Specifically, we modify the original data-generating process in Eq. 1 to incorporate the domain index $\mathbf{u}$ explicitly:

$$\mathbf{x} = g(\mathbf{z}, \epsilon), \quad \epsilon = e(\mathbf{z}, \mathbf{u}, \eta) \tag{5}$$

where $\mathbf{u}$ denotes the domain index, such that $\mathbf{u} \in [1, 2N + 1]$. Under these conditions, we establish the following identifiability theorem:

**Corollary 2.** *Suppose observational equivalence (Definition 1) holds between the true model $\{g, e, p(\mathbf{z}), p(\epsilon)\}$ and a learned model $\{\hat{g}, \hat{e}, p(\hat{\mathbf{z}})\}$, and the subspace identifiability condition in Theorem 1 is satisfied. Additionally, assume the following conditions:*

a *Latent variables are conditionally independent given domain $\mathbf{u}$: $p(\mathbf{z}|\mathbf{u}) = \prod_{n=1}^{N} p(\mathbf{z}^n|\mathbf{u})$;*

b *There exist $2N + 1$ distinct domain values $\mathbf{u} \in [1, 2N + 1]$, such that the $2N$ vectors $\mathbf{w}(\mathbf{z}, \mathbf{u}) - \mathbf{w}(\mathbf{z}, \mathbf{u}_0)$ (with $\mathbf{u} \ne \mathbf{u}_0$) are linearly independent, where the vector $\mathbf{w}(\mathbf{z}, \mathbf{u})$ is defined as:*

$$\mathbf{w}(\mathbf{z}, \mathbf{u}) = \{\mathbf{v}(\mathbf{z}, \mathbf{u}), \mathbf{v}'(\mathbf{z}, \mathbf{u})\}$$

*with*

$$\mathbf{v}(\mathbf{z}, \mathbf{u}) = \left( \frac{\partial \log p(\mathbf{z}^1|\mathbf{u})}{\partial \mathbf{z}^1}, \dots, \frac{\partial \log p(\mathbf{z}^N|\mathbf{u})}{\partial \mathbf{z}^N} \right)$$

$$\mathbf{v}'(\mathbf{z}, \mathbf{u}) = \left( \frac{\partial^2 \log p(\mathbf{z}^1|\mathbf{u})}{(\partial \mathbf{z}^1)^2}, \dots, \frac{\partial^2 \log p(\mathbf{z}^N|\mathbf{u})}{(\partial \mathbf{z}^N)^2} \right)$$

*Then $\{\hat{\mathbf{z}}^{\hat{n}} | \hat{n} \in [1, N]\}$ must be a component-wise transformation of a permuted version of true $\{\mathbf{z}^n | n \in [1, n]\}$*

**Proof Sketch.** See the Appendix B.4 for details. Theorem 1 gives an invertible $h$ with $\hat{\mathbf{z}} = h(\mathbf{z})$ and $\mathbf{z} = h^{-1}(\hat{\mathbf{z}})$. Using change of variables and Assumption a, we can obtain $\log p_{\hat{\mathbf{z}}|\mathbf{u}}(\hat{\mathbf{z}} \mid \mathbf{u}) = \sum_{i=1}^{n} \log p_{\mathbf{z}^i|\mathbf{u}}(\mathbf{z}^i \mid \mathbf{u}) + \log |\det J_{h^{-1}}(\hat{\mathbf{z}})|$. Taking second derivatives in $(\hat{\mathbf{z}}^k, \hat{\mathbf{z}}^v)$, $k \neq v$, the left-hand side vanishes while the right-hand side yields a linear system in $\{\tilde{h}^{i,(k)}\tilde{h}^{i,(v)}, \tilde{h}^{i,(k,v)\prime}\}_{i=1}^{n}$, where $\tilde{h}^{i,(k)} = \partial \mathbf{z}^i / \partial \hat{\mathbf{z}}^k$ and $\tilde{h}^{i,(k,v)\prime} = \partial^2 \mathbf{z}^i / (\partial \hat{\mathbf{z}}^k \partial \hat{\mathbf{z}}^v)$. By Assumption b, the resulting $2n$ coefficient vectors are linearly independent, forcing $\tilde{h}^{i,(k)}\tilde{h}^{i,(v)} = 0$ and $\tilde{h}^{i,(k,v)\prime} = 0$ for all $i$ and $k \neq v$. Thus each row and column of $J_{h^{-1}}$ has a single nonzero entry, which yields component-wise identifiability.

**Remark:** Distributional variability assumptions similar to Assumption b have been widely adopted in the literature on latent variable identifiability (Kong et al., 2022; Zhang et al., 2024). Intuitively, this assumption ensures that auxiliary variable $\mathbf{u}$ induces sufficient variability across latent dimensions. Building upon Theorem 1, our conclusion of Theorem 2 differs notably from previous works by explicitly allowing dependencies between $\mathbf{z}$ and $\epsilon$ in Eq. 5. In other words, our work degenerates to previous results without Theorem 1 and modeling the dependent noise $\epsilon$.

## 6 APPROACH

Building upon our established identifiability results, we now introduce an unsupervised method specifically designed for learning $\hat{\mathbf{z}}$. Our proposed method aims to achieve observational equivalence by explicitly modeling the data-generating process described in Eq. 1 Details of modeling Eq. 5 and the corresponding experiments are in Appendix D, respectively. In particular, we formulate the joint density corresponding to Eq. 1 as follows:

$$p(\mathbf{z}, \epsilon, \mathbf{x}) = p_\theta(\mathbf{x}|\mathbf{z}, \epsilon)p_\gamma(\epsilon|\mathbf{z})p_\delta(\mathbf{z}) \tag{6}$$

where parameters $\theta$ denotes the parameters of $g$. $\gamma$ denotes the parameters of $e$, and $\delta$ parameterizes $p(\mathbf{z})$. The second equation leverages the fact that $\mathbf{x}$ is independent of $\{\mathbf{z}, \epsilon\}$ given $\tilde{\mathbf{x}}$ in Eq. 1. To uncover the latent variables $\mathbf{z}$ and $\epsilon$ from observed data $\mathbf{x}$, we introduce two encoders, $q_\psi(\mathbf{z}|\mathbf{x})$ and $q_\phi(\epsilon|\mathbf{x})$ parameterized by $\psi$ and $\phi$, respectively.

To the end of learning Eq. 6, we bulid our approach upon the framework of Beta-VAE (Higgins et al., 2016). The overall architecture of our framework is illustrated in Figure 2. In what follows, we detail each part of our proposed model.

### 6.1 NETWORK DESIGN

Learning the log-likelihood in Eq. 6 via variational inference suggests the architecture for our approach composing of the following key elements. Specifically, the architecture includes two encoders: $q_\psi(\hat{\mathbf{z}}|\mathbf{x})$ for inferring latent variables $\mathbf{z}$, and $q_\phi(\hat{\epsilon}|\mathbf{x})$ for estimating the posterior of noise term $\epsilon$. These latent representations are then utilized by a decoder $p_\theta(\hat{\mathbf{x}}|\hat{\mathbf{z}}, \hat{\epsilon})$ to reconstruct the observations $\mathbf{x}$. Additionally, we regularize the latent variables by constraining their posterior distributions via the KL divergence to match the learned priors. We detail each of these modules below.

**Encoder** $q_\psi(\hat{\mathbf{z}}|\mathbf{x})$**:** We parameterize $q_\psi(\hat{\mathbf{z}}|\mathbf{x})$ as an isotropic Gaussian characterized by mean $\mu_{\mathbf{z}}$ and covariance $\sigma_{\mathbf{z}}$. To approximate this posterior, we employ a neural network encoder constructed with an MLP followed by a leaky ReLU activation:

$$\hat{\mathbf{z}} \sim \mathcal{N}(\mu_{\mathbf{z}}, \sigma_{\mathbf{z}}), \quad \mu_{\mathbf{z}}, \sigma_{\mathbf{z}} = \text{LeakyReLU}(\text{MLP}(\mathbf{x})) \tag{7}$$

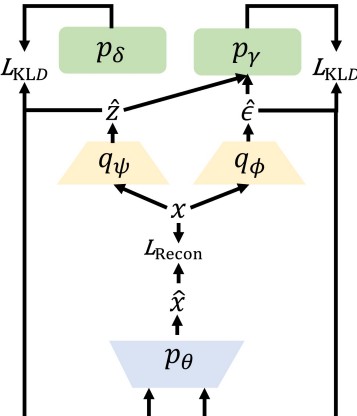

Figure 2: The overall framework of our proposed approach consists of: (1) two encoders $q_\psi$ and $q_\phi$ that map observations $\mathbf{x}_t$ to $\hat{\mathbf{z}}$ and $\hat{\epsilon}$, respectively; (2) a decoder that reconstructs observations $\hat{\mathbf{x}}$ from $\hat{\mathbf{z}}$ and $\hat{\epsilon}$; and (3) two prior estimation modules $p_\delta$ and $p_\gamma$ that models the prior of $\mathbf{z}$ and $\epsilon$, respectively. We train the framework by $L_{\text{Recon}}$ along with $L_{\text{KLD}}$.

**Encoder** $q_\phi(\hat\epsilon|\mathbf{x})$**:** Similarly, we parameterize $q_\phi(\hat\epsilon|\mathbf{x})$ as another isotropic Gaussian distribution:

$$\hat\epsilon \sim \mathcal{N}(\mu_\epsilon, \sigma_\epsilon), \quad \mu_\epsilon, \sigma_\epsilon = \text{LeakyReLU}(\text{MLP}(\mathbf{x})) \tag{8}$$

**Prior Estimation** $p_\delta(\mathbf{z})$**:** We estimate the prior $p_\delta(\mathbf{z})$ as a factorized Gaussian across latent dimensions, since we assume the independence of each dimension of $\mathbf{z}$ in Assumption A of Theorem 1:

$$p_\delta(\mathbf{z}) = \prod_{n=1}^{N} p_\delta(\mathbf{z}^n), \quad \mathbf{z}^n \sim \mathcal{N}(0, 1) \tag{9}$$

**Prior Estimation** $p_\gamma(\epsilon|\mathbf{z})$**:** Direct estimation of the arbitrary density $p_\gamma(\epsilon|\mathbf{z})$ poses a substantial challenge. To overcome this, we introduce a transformation-based module leveraging normalizing flows, representing the prior distribution as a Gaussian transformed via an invertible mapping. Suppose each component of $\epsilon$ is independent conditioning on $\mathbf{z}$, $\forall m \in [1, M]$, the prior model is formulated through: $\hat\eta^m = \hat{e}^{-1,m}(\hat\epsilon^m|\hat{\mathbf{z}})$. Using the change-of-variable, the prior distribution of $\hat\epsilon^m$ is computed as: $p_\gamma(\hat\epsilon^m|\hat{\mathbf{z}}) = p(\hat\eta^m)\left|\frac{\partial \hat{e}^{-1,m}}{\partial \hat\epsilon^m}\right| = p_\gamma(\hat{e}^{-1,m}(\hat\epsilon^m|\hat{\mathbf{z}}))\left|\frac{\partial \hat{e}^{-1,m}}{\partial \hat\epsilon^m}\right|$. Aggregating across all dimensions, the complete prior distribution is given by:

$$p_\gamma(\hat\epsilon|\hat{\mathbf{z}}) = \prod_{m=1}^{M} p(\hat\eta^m)\left|\frac{\partial \hat{e}^{-1,m}}{\partial \hat\epsilon^m}\right| \tag{10}$$

The normalizing flow transformation $\hat{e}$ is implemented using a stacked MLP.

**Decoder** $p_\theta(\hat{\mathbf{x}}|\hat{\mathbf{z}}, \hat\epsilon)$**:** The decoder generates the reconstructed observations $\hat{\mathbf{x}}$ from inferred latent variables $\hat{\mathbf{z}}$ and $\hat\epsilon$. It is implemented using an MLP followed by leaky ReLU activations:

$$\hat{\mathbf{x}} = \text{LeakyReLU}(\text{MLP}(\hat{\mathbf{z}}, \hat\epsilon)) \tag{11}$$

## 6.2 TRAINING OBJECTIVE

In this work, we extend the learning objective from the Beta-VAE framework (Higgins et al., 2016) by introducing a modified evidence lower bound (ELBO) The full ELBO objective is defined as:

$$\mathcal{L}_{\text{ELBO}} = \underbrace{\mathbb{E}_{\hat{\mathbf{z}}\sim q_\psi, \hat\epsilon\sim q_\phi}\left[\log p_\theta(\hat{\mathbf{x}}|\hat{\mathbf{z}}, \hat\epsilon)\right]}_{\mathcal{L}_{\text{Recon}}} + \underbrace{\lambda \left\|J_{\hat{g}}(\hat{\mathbf{z}})\right\|_1}_{\text{Sparsity Regularization}}$$
$$\underbrace{-\beta_1 \mathbb{E}_{\hat{\mathbf{z}}\sim q_\psi}\left(\log q(\hat{\mathbf{z}}|\mathbf{x}) - \log p_\delta(\mathbf{z})\right) - \beta_2 \mathbb{E}_{\hat{\mathbf{z}}\sim q_\psi, \hat\epsilon\sim q_\phi}\left(\log q(\hat\epsilon|\mathbf{x}) - \log p_\gamma(\hat\epsilon|\hat{\mathbf{z}})\right)}_{\mathcal{L}_{\text{KLD}}} \tag{12}$$

where $\lambda$, $\beta_1$ and $\beta_2$ are hyperparameters that balance the KL divergence penalties. The reconstruction term $\mathcal{L}_{\text{Recon}}$ measures the discrepancy between reconstructed observations $\hat{\mathbf{x}}$ and original inputs $\mathbf{x}$, implemented as a mean squared error loss. The KL divergence terms encourage the learned posterior distributions to match the assumed priors over $\mathbf{z}$ and $\epsilon$. Additionally, we regularize the decoder using the $\ell_1$ norm of the Jacobian matrix $J_{\hat{g}}(\hat{\mathbf{z}})$. This encourages the structural sparsity of learned $\hat{g}$. Following standard practice, we use the $\ell_1$ norm as a differentiable surrogate for $\ell_0$ sparsity constraints. Please refer to Appendix C for the details of network architectures.

# 7 EXPERIMENTS

## 7.1 SYNTHETIC EXPERIMENTS

**Experimental Setup** To thoroughly evaluate the capability of our approach in learning causal processes and accurately identifying latent variables, we perform simulation experiments using randomly generated causal structures with specified sample sizes and variable dimensions. Specifically, we create a synthetic dataset satisfying our data-generating process described in Eq. 1 (details in Appendix C.1).

For evaluation, we utilize the Mean Correlation Coefficient (MCC) as our primary metric, which quantifies the accuracy of latent variable recovery by computing the mean absolute correlation between the estimated and true latent variables. MCC scores range from 0 to 1, with higher values indicating better identifiability.

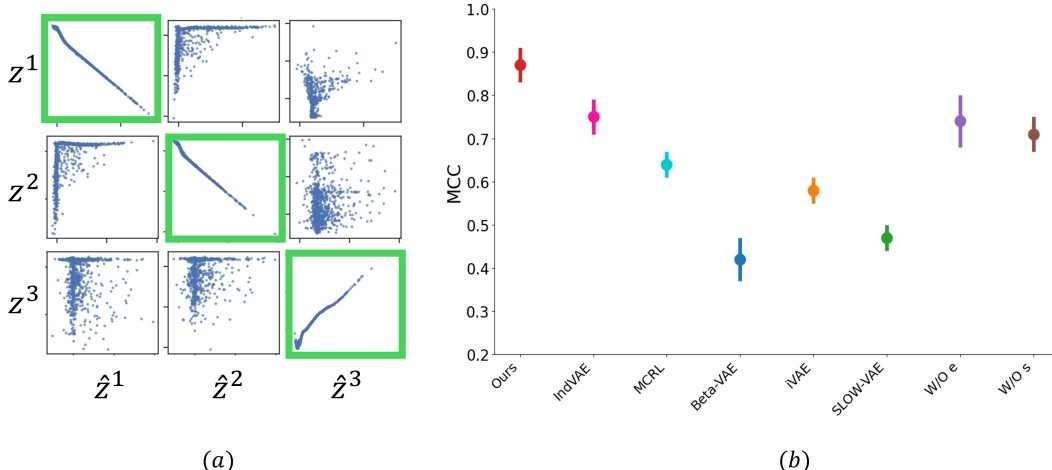

$(a)$ $(b)$

Figure 3: (a) Visualization of the correlations between each component of true latent variables ($\mathbf{z}^i$) and their corresponding component of estimated latent variables ($\hat{\mathbf{z}}^i$) using our approach. The green bounding boxes highlight the components that are identified. (b) Mean Correlation Coefficient (MCC) scores comparing our framework with state-of-the-art approaches, including IndVAE, MCRL, BetaVAE, iVAE, and SlowVAE, as well as the ablation baselines W/O $e$ and W/O $s$.

**Results** We evaluate our method against several state-of-the-art approaches for latent variable identification. Specifically, we compare our approach to the IndVAE that we build upon (Hu, 2008), which assumes conditional independence among observations given latent variables (see the details of objective in Sec. C.3). The Multimodal Causal Representation Learning framework (MCRL) proposed by (Sun et al., 2025) relies on the assumption that, the noise term $\epsilon$ is independent of the latent variables $\mathbf{z}$. Furthermore, we benchmark against classical representation learning methods such as Beta-VAE (Higgins et al., 2016), iVAE (Khemakhem et al., 2020), and SLOW-VAE (Klindt et al., 2020).

As illustrated in Figure 3(b), our method achieves the highest MCC score among all methods. We attribute this superior performance to ours capability to effectively handle generalized dependency structures by accurately disentangling latent variables $\mathbf{z}$ from the dependent noise term $\epsilon$. This fundamental advantage enables our approach to more precisely uncover the true underlying data-generating process. In addition to this quantitative result, Figure 3(a) provides a visual representation of the disentanglement between the true latent variables and their estimates.

**Ablation Study and Discussion** To elucidate the significance of key assumptions underlying our data generating process, we conduct an ablation study that specifically assesses the impact of generalized dependency structures and structural sparsity. We introduce two ablation baselines for comparison: (1) "W/O $e$", which removes $\epsilon = e(\mathbf{z})$ from Eq. 1. Accordingly, the likelihood becomes $p(\mathbf{z}, \epsilon, \mathbf{x}) = p(\mathbf{x}|\mathbf{z}, \epsilon)p(\mathbf{z}, \epsilon)$; (2) "W/O $s$", which drops the structural sparsity assumption imposed on $g$.

We present the results of this ablation study in Figure 3(b). Notably, our proposed method outperforms both the "W/O $e$" and "W/O $s$" baselines, highlighting the critical role of explicitly modeling dependent noise and structural sparsity. The substantial performance gap observed between our approach and the "W/O $e$" baseline tips the balance towards the necessity of modeling $\epsilon = e(\mathbf{z})$ to accurately capture the dependencies between $\epsilon$ and $\mathbf{z}$. Similarly, the diminished performance of the "W/O $s$" baseline emphasizes the essential contribution of structural sparsity within the mixing function $g$.

## 7.2 REAL-WORLD EXPERIMENT

**Task Setup:** To validate our proposed identifiability theories in realistic and complex scenarios, we apply them to the task of Person Index classification, a subtask of person ReID. In Person Index classification, the goal is to assign a unique identity index to each individual, based on input im-

ages. This setup aligns well with our generalized dependency structure setting, as each image of an individual inherently contains noise, such as varying poses, gaits, or clothes, making it challenging to disentangle these factors from the underlying identity. Also, different body parts cannot be independent conditioning upon the latent person identify index. Consequently, this task serves as a solid playground for evaluating the robustness and efficacy of our theoretical framework in addressing real-world complexities.

In our implementation, we first employ a pretrained feature extractor to derive feature representations of each individual person, denoted as $\mathbf{x} \in \mathbb{R}^K$. We consider $\mathbf{x}$ are generated by the latent variable $\mathbf{z} \in \mathbb{R}^N$, directly associated with each person's identity index, along with a dependent noise variable $\epsilon \in \mathbb{R}^M$, capturing variations such as pose, gait, or clothes. Inspired by the two-phase training pipeline proposed by (Li et al., 2024; 2025a), we adapt our approach to Person Index classification task as follows. First, we train our approach by optimizing the objective function detailed in Eq. 12. Subsequently, we introduce a classifier $\hat{c}$, implemented by a multilayer perceptron (MLP), to predict the one-hot encoded index label $\hat{y}$ from the inferred latent representation $\hat{\mathbf{z}}$: $\hat{y} = \text{MLP}(\hat{\mathbf{z}})$ The classifier is optimized using a cross-entropy loss given by: $\mathcal{L}_{\text{cls}}^{\text{CE}} = -\mathbb{E}_{\hat{y}}\left[\text{one-hot}(y) \cdot \log(\text{softmax}(\hat{y}))\right]$ where one-hot$(y)$ denotes the one-hot embedding of the true person index label. More data preprocessing details can be found in Appendix C.2.

**Data and Comparing Approaches** We conduct our experiments on the MSMT17 dataset (Wei et al., 2018), which comprises images of 4,101 unique individuals. Each individual in the dataset has more than 10 images, resulting in a total of over 120,000 images. We partition the dataset into three parts: 60% for training, 20% for validation, and the remaining 20% for test.

For performance comparison, we select several state-of-the-art methods on the task of Person Index classification, including GTL (Yang et al., 2025), AGW (Ye et al., 2021), TransReID (He et al., 2021), and CLIPReID (Li et al., 2023a). We also benchmark against MCRL (Sun et al., 2025) and IndVAE (Hu, 2008) to evaluate the efficacy of our identifiability results under generalized dependency structure.

Table 1: Comparison of Top-1 Accuracy on MSMT17 dataset

| Methods | Acc |
| --- | --- |
| AGW (Ye et al., 2021) | $85.5 \pm 1.2$ |
| TransReID (He et al., 2021) | $87.8 \pm 0.5$ |
| CLIPReID (Li et al., 2023a) | $90.1 \pm 0.3$ |
| GTL Yang et al. (2025) | $91.5 \pm 1.2$ |
| MCRL (Sun et al., 2025) | $92.6 \pm 0.9$ |
| IndVAE (Hu, 2008) | $93.1 \pm 0.5$ |
| Ours | $\mathbf{94.4 \pm 0.7}$ |

**Results & Discussions:** Table 1 reports the comparison of Top-1 Accuracy (Acc) among state-of-the-art methods on the MSMT17 dataset. Our method achieves a superior performance, substantially surpassing approaches that do not explicitly handle dependent noise, such as IndVAE (Hu, 2008) and MCRL (Sun et al., 2025). More specifically, our approach attains the highest accuracy of $94.4 \pm 0.7$, significantly improving upon the previous best performance of $93.1 \pm 0.5$ obtained by IndVAE. Furthermore, our proposed method demonstrates notable improvements over leading methods including GTL (Yang et al., 2025), CLIPReID (Li et al., 2023a), TransReID (He et al., 2021), and AGW (Ye et al., 2021), outperforming them by significant margins. These results clearly highlight the effectiveness and robustness of our proposed framework in accurately addressing generalized dependency structures in complex real-world scenarios.

## 8 CONCLUSION

This work introduces a set of novel identifiability guarantees under generalized dependency structures in which (i) observations can remain dependent given the latent variables and (ii) the noise may depend on the latents. Our theoretical framework establishes identifiability in two main steps. First, we rigorously prove the subspace identifiability by leveraging spectral decomposition techniques grounded in perturbation theory. Building upon this foundation, we further demonstrate component-wise identifiability. We validate our theoretical contributions through comprehensive experiments on both synthetic datasets and real-world tasks, showing the efficacy of our findings. While we have demonstrated the effectiveness of our approach on visual-based task, the lack of other applications is a limitation of this work.

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

## A    NOTIONS



*Table of notions*

| *Variables* | | | |
|---|---|---|---|
| $\mathbf{x} \in \mathbb{R}^K$ | Observations | $\hat{\mathbf{x}} \in \mathbb{R}^K$ | Reconstructions |
| $\kappa$ | The distance between eigenvalues | $\mathbf{u} \in [1, 2N+1]$ | Auxiliary domain variable |
| $\mathbf{z} \in \mathbb{R}^N$ | Latent variables | $\hat{\mathbf{z}} \in \mathbb{R}^N$ | Latent variable estimations |
| $\epsilon$ | True dependent noise term | $\hat{\epsilon}$ | Estimation of $\epsilon$ |
| $\eta$ | Auxiliary variable | $\hat{\eta}$ | Estimation of $\eta$ |
| *Indices* | | | |
| $\{\mathbf{a}, \mathbf{b}, \mathbf{c}\}$ | The indices of partitions of $\mathbf{x}$ | $\{A, B, C\}$ | The indices of partitions of $\mathbf{x}'$ |
| $n \in [N]$ | The indices of $\mathbf{z}$ | $\hat{n} \in [N]$ | The indices of $\hat{\mathbf{z}}$ |
| $i$ | The index of $\rho_\Lambda$ and $\rho$ | $j$ | The index of $\rho_\Lambda$ and $\rho$ |
| *Operators* | | | |
| $L$ | Integral linear operator | $\mathbf{d}$ | Difference operator |
| $\rho_\Lambda$ | The eigenvalue of $L_{\mathbf{x_a}|\mathbf{z}} L_{\mathbf{x_b}|\mathbf{z}} L_{\mathbf{x_a}|\mathbf{z}}^{-1}$ | $\rho$ | The eigenvalue of $L_{\mathbf{x_a}, \mathbf{x_b}|\mathbf{x_c}} L_{\mathbf{x_a}|\mathbf{x_c}}^{-1}$ |
| $Per$ | Perturbation operator | $\overline{Per}$ | The upper bound of $Per$ |
| *True & learned model* | | | |
| $g$ | True mixing function | $\hat{g}$ | Learned mixing function |
| $e$ | True function of $\epsilon$ | $\hat{e}$ | Learned function of $\epsilon$ |
| $J_g$ | The jacobian matrix of $g$ | $J_{\hat{g}}$ | The jacobian matrix of $\hat{g}$ |
| *Optimizations* | | | |
| $\psi$ | Parameters of posterior $q_\psi(\mathbf{z}|\mathbf{x})$ | $\phi$ | Parameters of posterior $q_\phi(\epsilon|\mathbf{x})$ |
| $\delta$ | Parameters of prior $p_\delta(\mathbf{z})$ | $\gamma$ | Parameters of prior $p_\gamma(\epsilon|\mathbf{z})$ |
| $\theta$ | Parameters of decoder $p_\theta(\mathbf{x}|\mathbf{z})$ | $|*|_1$ | $l_1$ norm on columns of $*$ |



## B    PROOF OF THEOREMS

### B.1    PREVIOUS RESULTS

In this section, we recapitulate and summarize the previous results from (Hu, 2008) in details.

*Consider observed variables* $\mathbf{x}' \in \mathbb{R}^K$ *and the estimated latent variables* $\hat{\mathbf{z}}' \in \mathbb{R}^N$, *suppose that there exist functions* $\hat{g}'$ *and* $\hat{e}'$ *satisfying the observational equivalence defined in Eq. 2, and the following assumptions hold:*

1. *For* $\mathbf{x}' = \{\mathbf{x}'_A, \mathbf{x}'_B, \mathbf{x}'_C\}$, $p(\mathbf{x}'|\mathbf{z}') = p(\mathbf{x}'_A|\mathbf{z}')p(\mathbf{x}'_B|\mathbf{z}')p(\mathbf{x}'_C|\mathbf{z}')$;

2. *The operators* $L_{\mathbf{x}'_A|\mathbf{z}'}$ *and* $L_{\mathbf{x}'_A|\mathbf{x}'_B}$ *are injective;*

3. $\forall \underline{\mathbf{z}}' \neq \bar{\mathbf{z}}', p(\mathbf{x}'_C; \underline{\mathbf{z}}') \neq p(\mathbf{x}'_C; \bar{\mathbf{z}}')$;

*then* $\mathbf{z}'$ *must be identified up to an invertible transformation* $h'$.

**Proof:** Given Assumption *1*, we can obtain the following:

$$p_{\mathbf{x}'_C \mathbf{x}'_A | \mathbf{x}'_B}(\mathbf{x}'_C, \mathbf{x}'_A | \mathbf{x}'_B) = \int p_{\mathbf{x}'_C \mathbf{x}'_A \mathbf{z}' | \mathbf{x}'_B}(\mathbf{x}'_C, \mathbf{x}'_A, \mathbf{z}' | \mathbf{x}'_B) d\mathbf{z}$$

$$= \int p_{\mathbf{x}'_C | \mathbf{x}'_A \mathbf{z}' \mathbf{x}'_B}(\mathbf{x}'_C | \mathbf{x}'_A, \mathbf{z}', \mathbf{x}'_B) p_{\mathbf{x}'_A \mathbf{z}' | \mathbf{x}'_B}(\mathbf{x}'_A, \mathbf{z}' | \mathbf{x}'_B) d\mathbf{z}$$

$$= \int p_{\mathbf{x}'_C | \mathbf{x}'_A \mathbf{z}'}(\mathbf{x}'_C | \mathbf{x}'_A, \mathbf{z}') p_{\mathbf{x}'_A \mathbf{z}' | \mathbf{x}'_B}(\mathbf{x}'_A, \mathbf{z}' | \mathbf{x}'_B) d\mathbf{z}$$

$$= \int p_{\mathbf{x}'_C | \mathbf{x}'_A \mathbf{z}'}(\mathbf{x}'_C | \mathbf{x}'_A, \mathbf{z}') p_{\mathbf{x}'_A | \mathbf{z}' \mathbf{x}'_B}(\mathbf{x}'_A | \mathbf{z}', \mathbf{x}'_B) p_{\mathbf{z}' | \mathbf{x}'_B}(\mathbf{z}' | \mathbf{x}'_B) d\mathbf{z}$$

$$= \int p_{\mathbf{x}'_C | \mathbf{x}'_A \mathbf{z}'}(\mathbf{x}'_C | \mathbf{x}'_A, \mathbf{z}') p_{\mathbf{x}'_A | \mathbf{z}'}(\mathbf{x}'_A | \mathbf{z}') p_{\mathbf{z}' | \mathbf{x}'_B}(\mathbf{z}' | \mathbf{x}'_B) d\mathbf{z}$$

$$= \int p_{\mathbf{x}'_C | \mathbf{z}'}(\mathbf{x}'_C | \mathbf{z}') p_{\mathbf{x}'_A | \mathbf{z}'}(\mathbf{x}'_A | \mathbf{z}') p_{\mathbf{z}' | \mathbf{x}'_B}(\mathbf{z}' | \mathbf{x}'_B) d\mathbf{z} \qquad (13)$$

Leveraging this eqation and Definition 4, we can derive the following operator:

$$(L_{\mathbf{x}'_C; \mathbf{x}'_A | \mathbf{x}'_B} f')(\mathbf{x}'_A) = \int \int p_{\mathbf{x}'_A | \mathbf{z}'}(\mathbf{x}'_A | \mathbf{z}') p_{\mathbf{x}'_C | \mathbf{z}'}(\mathbf{x}'_C | \mathbf{z}') p_{\mathbf{z}' | \mathbf{x}'_B}(\mathbf{z}' | \mathbf{x}'_B) f'(\mathbf{x}'_B) d\mathbf{x}'_B d\mathbf{z}$$

$$= \int p_{\mathbf{x}'_A | \mathbf{z}'}(\mathbf{x}'_A | \mathbf{z}') (L_{\mathbf{x}'_C; \mathbf{z}'} L_{\mathbf{z}' | \mathbf{x}'_B} f')(\mathbf{z}') d\mathbf{z}$$

$$= (L_{\mathbf{x}'_A | \mathbf{z}'} L_{\mathbf{x}'_C; Z} L_{\mathbf{z}' | \mathbf{x}'_B} f')(\mathbf{x}'_A) \qquad (14)$$

The above equation indicates:

$$L_{\mathbf{x}'_C; \mathbf{x}'_A | \mathbf{x}'_B} = L_{\mathbf{x}'_A | \mathbf{z}'} L_{\mathbf{x}'_C; \mathbf{z}'} L_{\mathbf{z}' | \mathbf{x}'_B} \qquad (15)$$

This equivalence holds over some functions space $\mathcal{G}(\mathcal{Z})$, given the factorization properties of the conditional densities established earlier.

Now, integrating over $\mathbf{x}'_C$, by using the fact that: $\int L_{\mathbf{x}'_C; \mathbf{x}'_A | \mathbf{x}'_B} f'(\mathbf{x}'_C) d\mathbf{x}'_C = L_{\mathbf{x}'_A | \mathbf{x}'_B} f'$ we can obtain:

$$L_{\mathbf{x}'_A | \mathbf{x}'_B} f'(\mathbf{x}'_A) = \int p_{\mathbf{x}'_A | \mathbf{x}'_B}(\mathbf{x}'_A | \mathbf{x}'_B) f'(\mathbf{x}'_B) d\mathbf{x}'_B$$

$$= \int \int p_{\mathbf{x}'_A | \mathbf{z}', \mathbf{x}'_B}(\mathbf{x}'_A | \mathbf{z}', \mathbf{x}'_B) p_{\mathbf{z}' | \mathbf{x}'_B}(\mathbf{z}' | \mathbf{x}'_B) f'(\mathbf{x}'_B) d\mathbf{x}'_B d\mathbf{z}$$

$$= \int p_{\mathbf{x}'_A | \mathbf{z}'}(\mathbf{x}'_A | \mathbf{z}') [L_{\mathbf{z}' | \mathbf{x}'_B} f'] d\mathbf{z}$$

$$= L_{\mathbf{x}'_A | \mathbf{z}'} L_{\mathbf{z}' | \mathbf{x}'_B} f'](\mathbf{x}'_A) \qquad (16)$$

where the second equation leverages $\mathbf{x}'_A \perp \mathbf{x}'_B \mid \mathbf{z}'$. By assuming the inejctivity of $L_{\mathbf{x}'_A | \mathbf{z}'}$ in Assumption *2*, we can arrive at $L_{Z | \mathbf{x}'_B} = L_{\mathbf{x}'_A | \mathbf{z}'}^{-1} L_{\mathbf{x}'_A | \mathbf{x}'_B}$. Substitute this to Eq. 15:

$$L_{\mathbf{x}'_C; \mathbf{x}'_A | \mathbf{x}'_B} L_{\mathbf{x}'_A | \mathbf{x}'_B}^{-1} = L_{\mathbf{x}'_A | \mathbf{z}'} L_{\mathbf{x}'_C; Z} L_{\mathbf{x}'_A | \mathbf{z}'}^{-1} \qquad (17)$$

where the LHS involves only observable variables, and the RHS explicitly depends on the latent variable $\mathbf{z}'$. $L_{\mathbf{x}'_C; \mathbf{z}'}$ determines the eigenvalues of $L_{\mathbf{x}'_A | Z} L_{\mathbf{x}'_C; \mathbf{z}'} L_{\mathbf{x}'_A | \mathbf{z}'}^{-1}$, whose diagonal entries correspond to the conditional distributions $p(\mathbf{x}'_C | \mathbf{z}')$. Each $\mathbf{z}'$ indexing a distinct conditional distribution of $p(\mathbf{x}'_C | \mathbf{z}')$. Under Assumption *3*, where $p(\mathbf{x}'_C \mid \mathbf{z}')$ are distinct for different values of $\mathbf{z}'$, the eigenvalues are distinct. This allows a bijective mapping $h' : \mathcal{Z} \to \mathcal{Z}$ to permute $\mathbf{z}'$ while preserving the values of $p(\mathbf{x}'_C \mid \mathbf{z}')$. Therefore, the latent variable can only be recovered up to such a permutation, i.e., $\hat{\mathbf{z}}' = h'(\mathbf{z}')$, which yields the identifiability up to an invertible transformation $h'$.

## B.2 PROOF OF THEOREM 1

In this section, we provide a formal proof of Theorem 1:

**Theorem 1** *Consider observed variables* $\mathbf{x} \in \mathbb{R}^K$ *and the estimated latent variables* $\hat{\mathbf{z}} \in \mathbb{R}^N$, *suppose that there exist functions* $\hat{g}$ *and* $\hat{e}$ *satisfying the observational equivalence defined in Eq. 2, and the following assumptions hold:*

   *i For* $\mathbf{x} = \{\mathbf{x_a}, \mathbf{x_b}, \mathbf{x_c}\}$, *we allow the dependencies such that* $p(\mathbf{x}|\mathbf{z}) \neq p(\mathbf{x_a}|\mathbf{z})p(\mathbf{x_b}|\mathbf{z})p(\mathbf{x_c}|\mathbf{z})$;

   *ii The operators* $L_{\mathbf{x_a}|\mathbf{z}}$, $L_{\mathbf{z}|\mathbf{x_c}}$, *and* $L_{\mathbf{x_a}|\mathbf{x_c}}$ *are injective;*

   *iii The operator* $L_{\mathbf{x_a},\mathbf{x_b}|\mathbf{x_c}} L_{\mathbf{x_a}|\mathbf{x_c}}^{-1}$ *has distinct eigenvalues with cardinality equal to that of* $L_{\mathbf{x_b}|\mathbf{z}}$;

   *iv* $L_{\mathbf{x_a}|\mathbf{z}} L_{\mathbf{x_b}|\mathbf{z}} L_{\mathbf{x_a}|\mathbf{z}}^{-1}$ *is self-adjoint.*

   *v* $\rho^i$ *denotes the* $i$-*th eigenvalue of the operator. Let* $\kappa = \min_{i \neq j} \frac{|\rho^i - \rho^j| - \alpha}{2} \geq 0$ *for some constant* $\alpha > 0$, *and* $\overline{|Per|} < \kappa$, *where* $\overline{|Per|}$ *denotes the upper bound of* $Per$;

   *vi There exists an operator* $M$ *such that* $M(L_{\mathbf{x_b}|\mathbf{z}}) = M(L_{\mathbf{x_b}|\tilde{h}(\mathbf{z})}) = t(\mathbf{z})$, *where* $t$ *is a differentiable transformation.*

*then for* $\tilde{h} \in \tilde{\mathcal{H}}$ *and* $t \in \mathcal{T}$ ($\tilde{\mathcal{H}} \cap \mathcal{T} \neq \emptyset$), *if* $h \in \tilde{\mathcal{H}} \cap \mathcal{T} \Rightarrow \hat{\mathbf{z}} = h(\mathbf{z}) = \tilde{h}(\mathbf{z}) = t(\mathbf{z})$. *In other words,* $\mathbf{z}$ *must be subspace identified.*

**Proof:** Suppose the observational equivalence (Definition 1) holds, our goal is to demonstrate subspace identifiability of $\mathbf{z}$ from the data generated process in Eq. 1. To such an end, we proceed in the following steps:

*Step 1: Operator Construction and Spectral Decomposition.*

Our approach follows the previous results in Sec. B.1, and proceeds to decompose the bounded linear operator $L_{\mathbf{x_a},\mathbf{x_b}|\mathbf{x_c}}$. Assumption i violates the conditional independence by $p(\mathbf{x}|\mathbf{z}) \neq p(\mathbf{x_a}|\mathbf{z})p(\mathbf{x_b}|\mathbf{z})p(\mathbf{x_c}|\mathbf{z})$, thus introduces a discrepancy between $L_{\mathbf{x_a},\mathbf{x_b}|\mathbf{x_c}}$ and $L_{\mathbf{x_a}|\mathbf{z}} L_{\mathbf{x_b}|\mathbf{z}} L_{\mathbf{z}|\mathbf{x_c}}$ according to Eq. 15. Consequently, we can define the difference operators:

$$\mathbf{d}(\mathbf{x_b}) = L_{\mathbf{x_a},\mathbf{x_b}|\mathbf{x_c}} - L_{\mathbf{x_a}|\mathbf{z}} L_{\mathbf{x_b}|\mathbf{z}} L_{\mathbf{z}|\mathbf{x_c}},$$
$$\mathbf{d} = \int \mathbf{d}(\mathbf{x_b}) \, d\mathbf{x_b} = L_{\mathbf{x_a}|\mathbf{x_c}} - L_{\mathbf{x_a}|\mathbf{z}} L_{\mathbf{z}|\mathbf{x_c}}. \tag{18}$$

Definition 4 guarantees boundedness of each term in Eq. 18, while Assumption ii ensures the existence of inverse operators $L_{\mathbf{x_a}|\mathbf{x_c}}^{-1}$, $L_{\mathbf{z}|\mathbf{x_c}}^{-1}$ and $L_{\mathbf{x_a}|\mathbf{z}}^{-1}$. Leveraging these, we rewrite:

$$L_{\mathbf{x_a}|\mathbf{z}} L_{\mathbf{x_b}|\mathbf{z}} L_{\mathbf{x_a}|\mathbf{z}}^{-1} = (L_{\mathbf{x_a},\mathbf{x_b}|\mathbf{x_c}} - \mathbf{d}(\mathbf{x_b}))(L_{\mathbf{x_a}|\mathbf{x_c}} - \mathbf{d})^{-1}$$
$$= L_{\mathbf{x_a},\mathbf{x_b}|\mathbf{x_c}} L_{\mathbf{x_a}|\mathbf{x_c}}^{-1} + Per, \tag{19}$$

where $Per$ represents the perturbation term arising from the violation of conditional independence.

*Step 2: Eigenvalue Uniqueness.*

By Assumption iii, the operator $L_{\mathbf{x_a},\mathbf{x_b}|\mathbf{x_c}} L_{\mathbf{x_a}|\mathbf{x_c}}^{-1}$ has distinct eigenvalues whose cardinality matches that of $L_{\mathbf{x_b}|\mathbf{z}}$. To establish the uniqueness of each eigenvalue in $L_{\mathbf{x_b}|\mathbf{z}}$ despite the perturbation, we apply Weyl's inequality (Kato, 2013) to Eq. 19 under Assumption iv. Let $\rho^i$ denote the $i$-th eigenvalue of $L_{\mathbf{x_a},\mathbf{x_b}|\mathbf{x_c}} L_{\mathbf{x_a}|\mathbf{x_c}}^{-1}$, and $\rho_\Lambda^i$ represent the corresponding eigenvalue in $L_{\mathbf{x_b}|\mathbf{z}}$:

$$|\rho_\Lambda^i - \rho^i| \leq \|Per\| \leq \overline{Per}, \tag{20}$$

where $\|\cdot\|$ is the $l_2$ operator norm, and $\overline{Per}$ denotes its upper bound. Starting from Eq. 20, we can obtain:

$$|\rho_\Lambda^i - \rho^i| \leq \overline{Per}, \quad \text{i.e.,} \quad \rho_\Lambda^i \in (\rho^i - \overline{Per}, \rho^i + \overline{Per}). \tag{21}$$

Suppose, for the sake of contradiction, two distinct eigenvalues, denoted $\rho_\Lambda^1$ and $\rho_\Lambda^2$, fall within the same interval $(\rho^i - \overline{Per}, \rho^i + \overline{Per})$. Assumption v further suggests that $\kappa = \min_{i \neq j} \frac{|\rho^i - \rho^j| - \alpha}{2} > 0$ and $\overline{Per} < \kappa$, we thus have:

$$|\rho_\Lambda^1 - \rho^i| \leq \overline{Per} < \frac{|\rho^i - \rho^j|}{2},$$
$$|\rho_\Lambda^2 - \rho^i| \leq \overline{Per} < \frac{|\rho^i - \rho^j|}{2}. \tag{22}$$

Applying the triangle inequality, we obtain:

$$|\rho_\Lambda^1 - \rho_\Lambda^2| \leq |\rho_\Lambda^1 - \rho^i| + |\rho_\Lambda^2 - \rho^i| < |\rho^i - \rho^j| = 2\kappa. \tag{23}$$

However, inequality 23 implies:

$$\frac{|\rho^1 - \rho^2|}{2} < \kappa, \tag{24}$$

which directly contradicts the definition of $\kappa$. Thus, each eigenvalue $\rho_\Lambda^i$ is unique within its respective interval.

The uniqueness of $\rho_\Lambda^i$ means that $L_{x_b|z}$ is determined by such decomposition. Specifically, permute $z$ introduces a permute operator $P$, such that, $PL_{x_b|z}P^{-1} = L_{x_b|\bar{h}(z)}$, where $\bar{h}$ denotes a permute transformation. Therefore, associated eigenfunctions $L_{x_a|z}$ and $L_{x_a|z}^{-1}$ are permuted, but the eigenvalues $L_{x_b|z}$ remains invariant. Since $\bar{h}$ is a permute transformation, it has to be invertible. We take inspirations from the conclusion from Sec. B.1, the distinct eigenvalues imply the existence of a re-labeling permutation $\tilde{z} = \tilde{h}(\mathbf{z})$, where $\tilde{h} : \mathcal{Z} \to \mathcal{Z}$ is bijective, and the eigenvalues $\rho_\Lambda$ remain invariant after such permuted transformation of $\mathbf{z}$.

*Step 3: Connecting Bijection $\tilde{h}$ with Differentiable Transformation $h$.*

Definition 2 requires the invertible transformation $h$ to be differentiable. However, the eigenvalue-based bijection $\tilde{h}$ alone may not satisfy this differentiability constraint. To resolve this, we invoke Assumption vi, which guarantees:

$$M(L_{\mathbf{x_b}|\mathbf{z}}) = M(L_{\mathbf{x_b}|\tilde{h}(\mathbf{z})}) = t(\mathbf{z}),$$

for a differentiable function $t$. Thus, defining the function classes $\tilde{\mathcal{H}}$ (bijections induced by eigenvalues) and $\mathcal{T}$ (differentiable transformations), if $h \in \tilde{\mathcal{H}} \cap \mathcal{T} \neq \emptyset$, the observational equivalence implies that $h$ coincides with $\tilde{h}$ and $t$, yielding: $\hat{\mathbf{z}} = h(\mathbf{z})$. Hence, we conclude the subspace identifiability of $\mathbf{z}$.

### B.3 PROOF OF COROLLARY 1

**Corollary 1** *Consider the true model $\{g, e, p(\mathbf{z}), p(\eta)\}$ and a learned model $\{\hat{g}, \hat{e}, p(\hat{\mathbf{z}}), p(\hat{\eta})\}$ that satisfy observational equivalence (Definition 1) and subspace identifiability (Theorem 1). Suppose further the following assumptions and regularization conditions hold:*

A *Latent dimensions of $\mathbf{z}$ are independent: $p(\mathbf{z}) = \prod_{n=1}^N p(\mathbf{z}^n)$;*

B *For each dimension $n \in [1, N]$ of $\mathbf{z}$, there exist $\{\mathbf{z}^l\}_{l=1}^{|G^{n,:}|}$ such that:*

$$span\{J_g(\mathbf{z}^l)_{n,:}\}_{l=1}^{|G^{n,:}|} = \mathbb{R}_{G^{n,:}}^N, \quad and \quad [J_{\hat{g}}(\hat{\mathbf{z}}^l)_{n,:}]_{l=1}^{|\hat{G}^{n,:}|} \in \mathbb{R}_{\hat{G}^{n,:}}^N$$

C *For each $n \in [1, N]$, there exists a subset of indices $\mathcal{C}_k$ satisfying $\bigcap_{m \in \mathcal{C}_k} G^{m,:} = \{n\}$;*

D *Sparsity regularization: $|\hat{G}| \leq |G|$*

*Then, $\hat{\mathbf{z}}$ must correspond component-wise to a permutation of the true latent variables $\mathbf{z}$.*

**Proof:** Theorem 1 guarantees the existence of an invertible trasnformation $h$ such that $\hat{\mathbf{z}} = h(\mathbf{z})$ and, since the observational equivalence in Definition 1 indicates $\mathbf{x} = g(\mathbf{z}) = \hat{g}(\hat{\mathbf{z}})$, the chain rule yields

$$J_g(\mathbf{z}) = J_{\hat{g}}(\hat{\mathbf{z}}) J_h(\mathbf{z}) \tag{25}$$

Our goal is to show that $h$ is a composition of a permutation and component-wise diagonal transformations.

Let us denote $J_h$ by $\mathbf{H}$. According to our assumption, for each index $i$, the set of basis vectors $e \in \{J_g(\mathbf{z}^{(l)})_{i,:}\}_{l=1}^{|G_{i,:}|}$ spans the space $\mathbb{R}_{G_{i,:}}^n$. This means any vector in $\mathbb{R}_{G_{i,:}}^n$ can be expressed as a

linear combination of these basis vectors. In particular, Assumption B suggests that, for any standard basis vector $e_{j_0}$ with $j_0 \in G_{i,:}$ we have

$$e_{j_0} \mathbf{H} \in \mathbb{R}^n_{\hat{G}_{i,:}} \qquad \Longrightarrow \qquad \mathbf{H}_{j_0,:} \in \mathbb{R}^n_{\hat{G}_{i,:}}, \tag{26}$$

and therefore

$$\forall (i,j) \in G, \qquad \{i\} \times \operatorname{supp}(\mathbf{H}_{j,:}) \subset \hat{G}. \tag{27}$$

Because $J_g(\mathbf{z})$ and $J_{\hat{g}}(\hat{\mathbf{z}})$ both have full column rank $n$, $\mathbf{H}$ is invertible. By the Leibniz formula, there exists a permutation $\sigma$ with $\mathbf{H}_{i,\sigma(i)} \neq 0$ for all $i$, i.e., $\sigma(j) \in \operatorname{supp}(\mathbf{H}_{j,:})$ for all $j$. Combining this with equation 27 gives

$$\forall (i,j) \in G, \qquad (i, \sigma(j)) \in \hat{G}. \tag{28}$$

Define the permuted edge set $\sigma(G) = \{(i, \sigma(j)) : (i,j) \in G\}$. Then $\sigma(G) \subset \hat{G}$. Sparsity regularization D on the estimated Jacobian ensures $|\hat{G}| \leq |G| = |\sigma(G)|$, hence

$$\sigma(G) = \hat{G}. \tag{29}$$

Suppose, for contradiction, that $\mathbf{H}(z)$ is not a composition of a diagonal matrix and a permutation matrix, i.e., there exist $j_1 \neq j_2$ such that:

$$\operatorname{supp}(\mathbf{H}_{j_1,:}) \cap \operatorname{supp}(\mathbf{H}_{j_2,:}) \neq \emptyset. \tag{30}$$

Let $j_3$ be an element in this intersection, so $\sigma(j_3) \in \operatorname{supp}(\mathbf{H}_{j_1,:}) \cap \operatorname{supp}(\mathbf{H}_{j_2,:})$. Without loss of generality, assume $j_3 \neq j_1$. According to Assumption C, there exists a set $\mathcal{C}_{j_1}$ containing $j_1$ such that:

$$\bigcap_{i \in \mathcal{C}_{j_1}} G_{i,:} = \{j_1\}. \tag{31}$$

Since $j_3 \neq j_1$, it must be that:

$$j_3 \notin \bigcap_{i \in \mathcal{C}_{j_1}} G_{i,:}, \tag{32}$$

implying there exists some $i_3 \in \mathcal{C}_{j_1}$ such that:

$$j_3 \notin G_{i_3,:}. \tag{33}$$

However, since $j_1 \in G_{i_3,:}$, we have $(i_3, j_1) \in G$. Using Eq. 28, we find:

$$(i_3, \sigma(j_3)) \in \hat{G}. \tag{34}$$

But from Eq. 29, this means $(i_3, j_3) \in G$, which contradicts Eq. 33. This contradiction implies our assumption is false, and therefore $\mathbf{H}$ must be a composition of a permutation matrix and a diagonal matrix.

Together with the equation $J_g = J_{\hat{g}}\mathbf{H}$, we achieve the desired result that $t$ is composed of a permutation and component-wise invertible functions.

### B.4 PROOF OF COROLLARY 2

*Corollary 2 Suppose observational equivalence (Definition 1) holds between the true model $\{g, e, p(\mathbf{z})\}$ and the learned model $\{\hat{g}, \hat{e}, p(\hat{\mathbf{z}})\}$, and the subspace identifiability condition in Theorem 1 is satisfied. Additionally, assume the following conditions:*

*a Latent variables are conditionally independent given domain $\mathbf{u}$: $p(\mathbf{z}|\mathbf{u}) = \prod_{n=1}^{N} p(\mathbf{z}^n|\mathbf{u})$;*

*b There exist $2N + 1$ distinct domain values $\mathbf{u} \in [1, 2N+1]$, such that the $2N$ vectors $\mathbf{w}(\mathbf{z}, \mathbf{u}) - \mathbf{w}(\mathbf{z}, \mathbf{u}_0)$ (with $\mathbf{u} \neq \mathbf{u}_0$) are linearly independent, where the vector $\mathbf{w}(\mathbf{z}, \mathbf{u})$ is defined as:*

$$\mathbf{w}(\mathbf{z}, \mathbf{u}) = \{\mathbf{v}(\mathbf{z}, \mathbf{u}), \mathbf{v}'(\mathbf{z}, \mathbf{u})\}$$

*with*

$$\mathbf{v}(\mathbf{z}, \mathbf{u}) = \left( \frac{\partial \log p(\mathbf{z}^1|\mathbf{u})}{\partial \mathbf{z}^1}, \dots, \frac{\partial \log p(\mathbf{z}^N|\mathbf{u})}{\partial \mathbf{z}^N} \right)$$

$$\mathbf{v}'(\mathbf{z}, \mathbf{u}) = \left( \frac{\partial^2 \log p(\mathbf{z}^1|\mathbf{u})}{(\partial \mathbf{z}^1)^2}, \dots, \frac{\partial^2 \log p(\mathbf{z}^N|\mathbf{u})}{(\partial \mathbf{z}^N)^2} \right)$$

*Then $\{\hat{\mathbf{z}}^{\hat{n}} | \hat{n} \in [1, N]\}$ must be a component-wise transformation of a permuted version of true $\{\mathbf{z}^n | n \in [1, n]\}$*

**Proof:** By Theorem 1 there exists an invertible reparameterization $h : \mathcal{Z} \to \mathcal{Z}$ such that $\hat{\mathbf{z}} = h(\mathbf{z})$ and $\mathbf{z} = h^{-1}(\hat{\mathbf{z}})$. Applying the change-of-variables formula to the conditional densities (for any fixed $\mathbf{u}$) gives:

$$p_{\hat{\mathbf{z}}|\mathbf{u}}(\hat{\mathbf{z}} \mid \mathbf{u}) = p_{\mathbf{z}|\mathbf{u}}\big(h^{-1}(\hat{\mathbf{z}}) \mid \mathbf{u}\big) \left|\det J_{h^{-1}}(\hat{\mathbf{z}})\right|. \tag{35}$$

Taking logarithms yields

$$\log p_{\hat{\mathbf{z}}|\mathbf{u}}(\hat{\mathbf{z}} \mid \mathbf{u}) = \log p_{\mathbf{z}|\mathbf{u}}(\mathbf{z} \mid \mathbf{u}) + \log\left|\det J_{h^{-1}}(\hat{\mathbf{z}})\right|, \tag{36}$$

Under Assumption a, we have

$$\sum_{i=1}^{n} \log p_{\hat{\mathbf{z}}^i|\mathbf{u}}(\hat{\mathbf{z}}^i \mid \mathbf{u}) = \sum_{i=1}^{n} \log p_{\mathbf{z}^i|\mathbf{u}}(\mathbf{z}^i \mid \mathbf{u}) + \log\left|\det J_{h^{-1}}(\hat{\mathbf{z}})\right|. \tag{37}$$

Following Hyvärinen et al. (2024), take second derivatives with respect to $\hat{\mathbf{z}}^k$ and $\hat{\mathbf{z}}^v$ for $k \neq v$. Since each term on the left-hand side of Eq. 37 depends only on a single coordinate $\hat{\mathbf{z}}^i$, we have $\partial \log p_{\hat{\mathbf{z}}^i|\mathbf{u}}(\hat{\mathbf{z}}^i \mid \mathbf{u})/\partial \hat{\mathbf{z}}^k = 0$ for $i \neq k$, which implies

$$\frac{\partial^2}{\partial \hat{\mathbf{z}}^k \partial \hat{\mathbf{z}}^v} \sum_{i=1}^{n} \log p_{\hat{\mathbf{z}}^i|\mathbf{u}}(\hat{\mathbf{z}}^i \mid \mathbf{u}) = 0. \tag{38}$$

For the right-hand side, define for $i = 1, \dots, n$

$$\tilde{h}^{i,(k)} := \frac{\partial \mathbf{z}^i}{\partial \hat{\mathbf{z}}^k}, \qquad\qquad \tilde{h}^{i,(k,v)'} := \frac{\partial^2 \mathbf{z}^i}{\partial \hat{\mathbf{z}}^k \partial \hat{\mathbf{z}}^v}, \tag{39}$$

$$\eta_i'(\mathbf{z}^i, \mathbf{u}) := \frac{\partial}{\partial \mathbf{z}^i} \log p_{\mathbf{z}^i|\mathbf{u}}(\mathbf{z}^i \mid \mathbf{u}), \qquad \eta_i''(\mathbf{z}^i, \mathbf{u}) := \frac{\partial^2}{\partial \mathbf{z}^{i2}} \log p_{\mathbf{z}^i|\mathbf{u}}(\mathbf{z}^i \mid \mathbf{u}) \tag{40}$$

A direct application of the chain rule gives

$$\sum_{i=1}^{n} \left( \eta_i''(\mathbf{z}^i, \mathbf{u}) \, \tilde{h}^{i,(k)} \, \tilde{h}^{i,(v)} + \eta_i'(\mathbf{z}^i, \mathbf{u}) \tilde{h}^{i,(k,v)'} \right) + \frac{\partial^2}{\partial \hat{\mathbf{z}}^k \partial \hat{\mathbf{z}}^v} \log\left|\det J_{h^{-1}}(\hat{\mathbf{z}})\right| = 0. \tag{41}$$

Fix $(k, v)$ with $k \neq v$ and evaluate this identity at $2n+1$ distinct values of the conditioning variable, $\mathbf{u}^{(j)}$ for $j \in \{0, 1, \dots, 2n\}$. Subtracting the equation at $\mathbf{u}^{(0)}$ from that at $\mathbf{u}^{(j)}$ cancels the log-determinant term (which does not depend on $\mathbf{u}$) and yields, for $j = 1, \dots, 2n$,

$$\sum_{i=1}^{n} \left( [\eta_i''(\mathbf{z}^i, \mathbf{u}^{(j)}) - \eta_i''(\mathbf{z}^i, \mathbf{u}^{(0)})] \tilde{h}^{i,(k)} \, \tilde{h}^{i,(v)} + [\eta_i'(\mathbf{z}^i, \mathbf{u}^{(j)}) - \eta_i'(\mathbf{z}^i, \mathbf{u}^{(0)})] \tilde{h}^{i,(k,v)'} \right) = 0. \tag{42}$$

Let

$$\mathbf{w}(\mathbf{z}, \mathbf{u}) := \big( \eta_1''(\mathbf{z}^1, \mathbf{u}), \dots, \eta_n''(\mathbf{z}^n, \mathbf{u}), \, \eta_1'(\mathbf{z}^1, \mathbf{u}), \dots, \eta_n'(\mathbf{z}^n, \mathbf{u}) \big)^\top. \tag{43}$$

Under Assumption b, the $2n$ vectors $\mathbf{w}(\mathbf{z}, \mathbf{u}^{(j)}) - \mathbf{w}(\mathbf{z}, \mathbf{u}^{(0)})$ for $j = 1, \dots, 2n$ are linearly independent, so the only solution to the homogeneous linear system Eq. 42 is

$$\tilde{h}^{i,(k)} \, \tilde{h}^{i,(v)} = 0 \quad \text{and} \quad \tilde{h}^{i,(k,v)'} = 0 \qquad \text{for all } i \in \{1, \dots, n\} \text{ and all } k \neq v. \tag{44}$$

Hence each row of the Jacobian $J_{h^{-1}}(\hat{\mathbf{z}})$ has at most one nonzero entry, and all mixed second derivatives vanish. Since $h^{-1}$ is invertible, each row must in fact have exactly one nonzero entry; moreover, two distinct rows cannot share the same nonzero column (otherwise $\det J_{h^{-1}}(\hat{\mathbf{z}}) = 0$), so there exists a permutation $\pi$ such that

$$\hat{\mathbf{z}}^{\pi(i)} = h^i(\mathbf{z}^i) \qquad \text{for } i = 1, \dots, n, \tag{45}$$

which shows that $\hat{\mathbf{z}}$ is obtained from $\mathbf{z}$ by a permutation of component-wise invertible transformations.

## B.5   Identifying $\epsilon$

***Corollary*** *Consider the true model $\{g, e, p(\mathbf{z}), p(\eta)\}$ and a learned model $\{\hat{g}, \hat{e}, p(\hat{\mathbf{z}}), p(\hat{\eta})\}$ that satisfy observational equivalence (Definition 1) and subspace identifiability (Theorem 1). Further, the following assumptions also hold:*

1. *There exists an smmoth, invertible transformation $f$ between $z, \epsilon$ and $\hat{z}, \hat{\epsilon}$. Also, the diagonal entries of the jacobian of $f$ coincides with $\frac{\partial \hat{z}}{\partial z}$ and $\frac{\partial \hat{\epsilon}}{\partial \epsilon}$, respectively.*

2. *For $\mathbf{x} = \{\mathbf{x_a}, \mathbf{x_b}, \mathbf{x_c}\}$, we allow the dependencies such that $p(\mathbf{x}|\epsilon) \neq p(\mathbf{x_a}|\epsilon)p(\mathbf{x_b}|\epsilon)p(\mathbf{x_c}|\epsilon)$;*

3. *The operators $L_{\mathbf{x_a}|\epsilon}$, $L_{\epsilon|\mathbf{x_c}}$, and $L_{\mathbf{x_a}|\mathbf{x_c}}$ are injective;*

4. *The operator $L_{\mathbf{x_a},\mathbf{x_b}|\mathbf{x_c}}L_{\mathbf{x_a}|\mathbf{x_c}}^{-1}$ has distinct eigenvalues with cardinality equal to that of $L_{\mathbf{x_b}|\epsilon}$;*

5. *$L_{\mathbf{x_a}|\epsilon}L_{\mathbf{x_b}|\epsilon}L_{\mathbf{x_a}|\epsilon}^{-1}$ is self-adjoint.*

6. *$\rho^i$ denotes the $i$-th eigenvalue of the operator $L_{\mathbf{x_a},\mathbf{x_b}|\mathbf{x_c}}L_{\mathbf{x_a}|\mathbf{x_c}}^{-1}$. Let $\kappa = \min_{i \neq j} \frac{|\rho^i - \rho^j| - \alpha}{2} \geq 0$ for some constant $\alpha > 0$, and $\overline{|Per|} < \kappa$, where $\overline{|Per|}$ denotes the upper bound of $Per$;*

7. *There exists an operator $M_\epsilon$ such that $M_\epsilon(L_{\mathbf{x_b}|\epsilon}) = M_\epsilon(L_{\mathbf{x_b}|\tilde{h}_\epsilon(\epsilon)}) = t_\epsilon(\epsilon)$, where $t_\epsilon$ is a differentiable transformation.*

*then for $\tilde{h}_\epsilon \in \tilde{\mathcal{H}}_\epsilon$ and $t_\epsilon \in \mathcal{T}_\epsilon$ (where $\tilde{\mathcal{H}}_\epsilon$ and $\mathcal{T}_\epsilon$ are function classes, and $\tilde{\mathcal{H}}_\epsilon \cap \mathcal{T}_\epsilon \neq \emptyset$), if $h_\epsilon \in \tilde{\mathcal{H}}_\epsilon \cap \mathcal{T}_\epsilon \Rightarrow \hat{\epsilon} = h_\epsilon(\epsilon) = \tilde{h}_\epsilon(\epsilon) = t_\epsilon(\epsilon)$. In other words, $\epsilon$ must be subspace identified. Combinting with the conclusion from Themorem 1, we can further obtain the block-wise identifiability of $z$ and $\epsilon$.*

**Proof:** We can arrive at $\hat{\epsilon} = h_\epsilon(\epsilon)$ by following the same proof strategy as in Sec. B.2. Since we assume there exists an smooth, invertible transformation $f$ between $(z, \epsilon)$ and $(\hat{z}, \hat{\epsilon})$:

$$(\hat{z}, \hat{\epsilon}) = f(z, \epsilon). \tag{46}$$

The Jacobian of $f$ with respect to $(z, \epsilon)$ is

$$J_f = \begin{pmatrix} \dfrac{\partial \hat{z}}{\partial z} & \dfrac{\partial \hat{z}}{\partial \epsilon} \\ \dfrac{\partial \hat{\epsilon}}{\partial z} & \dfrac{\partial \hat{\epsilon}}{\partial \epsilon} \end{pmatrix}. \tag{47}$$

Theorem 1 gives $\hat{z} = h(z)$, hence $\partial \hat{z}/\partial \epsilon = 0$. Analogously, $\hat{\epsilon} = h_\epsilon(\epsilon)$ implies $\partial \hat{\epsilon}/\partial z = 0$. Therefore the Jacobian reduces to

$$J_f = \begin{pmatrix} \dfrac{\partial \hat{z}}{\partial z} & 0 \\ 0 & \dfrac{\partial \hat{\epsilon}}{\partial \epsilon} \end{pmatrix}, \tag{48}$$

which is block-diagonal. This shows that the transformation between $(z, \epsilon)$ and $(\hat{z}, \hat{\epsilon})$ is block-wise, i.e., we have block-wise identifiability of $z$ and $\epsilon$.

## C IMPLEMENTATION DETAILS

### C.1 SYNTHETIC EXPERIMENT

**Synthetic Data Generation Process:** Our data generating process for the synthetic experiment is as follows:

$$\epsilon_1 = \mathcal{N}(0, (\mathbf{z}_1)^2), \quad \tilde{g}_1(\mathbf{z}_1, \epsilon_1) = sinh(\mathbf{z}_1) \times \epsilon_1$$

$$\epsilon_2 = tanh(\mathbf{z}_2), \quad \tilde{g}_2(\mathbf{z}_2, \epsilon_2) = \frac{1}{1 + exp(\mathbf{z}_2)} \times \epsilon_2$$

$$\epsilon_3 = Laplace(0, |\mathbf{z}_3|), \quad \tilde{g}_3(\mathbf{z}_3, \epsilon_3) = (\mathbf{z}_3)^2 \times \epsilon_3$$

$$\tilde{\mathbf{x}}_m = \tilde{g}_m(\mathbf{z}_m, \epsilon_m), \quad \mathbf{x} = g(\tilde{\mathbf{x}}) = \sigma(R\tilde{\mathbf{x}}) \tag{49}$$

where $m \in [1, 3]$. We sample 10,000 states drawn from $\mathbf{z} \sim U(0, 1)$, $R \in \mathbb{R}^{3 \times 3}$ is a fixed full-rank matrix with sparse, small nonzero off-diagonal entries. $\tilde{\mathbf{x}} = (\tilde{\mathbf{x}}_1, \tilde{\mathbf{x}}_2, \tilde{\mathbf{x}}_3)^\top$, and $\sigma$ is a smooth strictly monotone scalar nonlinearity applied coordinate-wise. In practice, we randomly assign the roles of $\mathbf{x}_a, \mathbf{x}_b, \mathbf{x}_c$ to $\mathbf{x}_1, \mathbf{x}_2, \mathbf{x}_3$, so that $(\mathbf{x}_a, \mathbf{x}_b, \mathbf{x}_c)$ can be any random permutation of $(\mathbf{x}_1, \mathbf{x}_2, \mathbf{x}_3)$.

Table 2: The details of our network architectures for the experiment on MSTM-17 dataset, where BS means batch size, $N = 32$, $M = 32$ and $K = 1280$.

| Configuration | Description | Output dimensions |
|---|---|---|
| Encoder $q\psi$ | | |
| Input: $\mathbf{x}$ | | BS $\times K$ |
| Dense | 128 neurons, LeakyReLU | BS $\times 128$ |
| Dense | 128 neurons, LeakyReLU | BS $\times 128$ |
| Dense | Output embeddings | BS $\times 2N$ |
| Bottleneck | Compute mean and variance of posterior | $\mu_{\mathbf{z}}, \sigma_{\mathbf{z}}$ |
| Reparameterization | Sequential sampling | $\hat{\mathbf{z}}$ |
| | | |
| Encoder $q_\phi$ | | |
| Input: $\mathbf{x}$ | | BS $\times K$ |
| Dense | 128 neurons, LeakyReLU | BS $\times 128$ |
| Dense | 128 neurons, LeakyReLU | BS $\times 128$ |
| Dense | Output embeddings | BS $\times 2M$ |
| Bottleneck | Compute mean and variance of posterior | $\mu_\epsilon, \sigma_\epsilon$ |
| Reparameterization | Sequential sampling | $\hat{\epsilon}$ |
| | | |
| Decoder | | |
| Input: $\hat{\mathbf{z}}, \hat{\epsilon}$ | | BS $\times (N + M)$ |
| Dense | 128 neurons, LeakyReLU | BS $\times 128$ |
| Dense | 128 neurons, LeakyReLU | BS $\times 128$ |
| Dense | input embeddings | BS $\times K$ |
| | | |
| Prior module | | |
| Input | $\hat{\mathbf{z}}, \hat{\epsilon}$ | BS $\times (N + M)$ |
| InverseTransformation | $\hat{\eta}$ | BS $\times M$ |
| JacobianCompute | $\log |\det J_{\hat{e}}|$ | BS |
| | | |
| Classifier | | |
| Input: $\hat{\mathbf{z}}$ | | BS $\times N$ |
| Dense | 256 neurons, LeakyReLU | BS $\times 256$ |
| Dense | 256 neurons, LeakyReLU | BS $\times 256$ |
| Dense | output one-hot embeddings | BS $\times 4101$ |

**Implementations & Training Details.** In our synthetic experiments, we set the dimensions $N = 3$ and $K = 3$. The encoders, decoder, and normalizing flow modules were each implemented using single-layer multilayer perceptrons (MLPs) followed by Leaky ReLU activations.

Our implementation utilized PyTorch 1.11.0. For optimization, we adopted the AdamW optimizer Loshchilov & Hutter (2019), which is known for enhancing generalization in deep learning models.

The hyperparameters were configured as follows: a learning rate of $1 \times 10^{-3}$ and a batch size of 64. To guarantee robustness and statistical reliability, each model was trained using 10 different random seeds. We report the overall performance as the mean $\pm$ standard deviation computed across these runs. The loss function employed balances the reconstruction error and the KL-divergence, with weighting coefficients set to $\beta_1 = \beta_2 = 0.02$. All experiments were performed on a single NVIDIA GeForce RTX 2080 Ti GPU equipped with 11GB of memory.

## C.2 REAL-WORLD EXPERIMENT

To obtain a fair comparison, we adopt the approach outlined by Yang et al. (2025), employing the pretrained CLIP model (Radford et al., 2021) as the visual encoder to generate 1280-dimensional representations for $\mathbf{x}$. Table 2 summarizes the specific network architectures implemented for our experiments on the real-world MSTM17 dataset.

To train our framework, we utilize the AdamW optimizer combined with a cosine annealing learning rate schedule. The initial learning rate is set to $2 \times 10^{-3}$, with a weight decay parameter of $1 \times 10^{-2}$ to prevent overfitting. The ELBO loss function incorporates equal weighting coefficients $\beta_1 = \beta_2 = 0.02$. We use a batch size of 128, chosen to balance computational efficiency with optimization stability. The framework is implemented in PyTorch. Training is done for 80 epochs on a multi-GPU configuration comprising four NVIDIA GeForce RTX 2080 Ti GPUs, collectively providing 44GB of memory.

## C.3 THE OBJECTIVE OF INDVAE

In this section, we explain our trained objective for IndVAE, which is designed by taking inspirations from (Hu, 2008) To incorporate with the conditional independence assumption, we consider the $k$-th observed variable $\mathbf{x}^k$ is generated as $\mathbf{x}^k = g^k(\mathbf{z}, \epsilon)$. Accordingly, the log-likelihood of the data generating process of Eq. 1 can be transformed as follows:

$$
\begin{aligned}
\log p(\mathbf{z}, \epsilon, \mathbf{x}) &= \log p_\theta(\mathbf{x}|\mathbf{z}, \epsilon) + \log p_\gamma(\epsilon|\mathbf{z}) + \log p_\delta(\mathbf{z}) \\
&= \sum_{k=1}^{K} \log p_\theta(\mathbf{x}^k|\mathbf{z}, \epsilon) + \log p_\gamma(\epsilon|\mathbf{z}) + \log p_\delta(\mathbf{z})
\end{aligned}
\tag{50}
$$

Accordingly, the loss function becomes:

$$
\mathcal{L}_{\text{ELBO}} = \underbrace{\mathbb{E}_{\hat{\mathbf{z}} \sim q_\psi, \, \hat{\epsilon} \sim q_\phi} \left[ \sum_{k=1}^{K} \log p_\theta(\hat{\mathbf{x}}^k|\hat{\mathbf{z}}, \hat{\epsilon}) \right]}_{\mathcal{L}_{\text{Recon}}} + \underbrace{\|J_{\hat{g}}(\hat{\mathbf{z}})\|_1}_{\text{Sparsity Regularization}}
$$

$$
\underbrace{-\beta_1 \mathbb{E}_{\hat{\mathbf{z}} \sim q_\psi} \big( \log q\,(\hat{\mathbf{z}}|\mathbf{x}) - \log p_\delta(\mathbf{z}) \big) - \beta_2 \mathbb{E}_{\hat{\mathbf{z}} \sim q_\psi, \hat{\epsilon} \sim q_\phi} \big( \log q\,(\hat{\epsilon}|\mathbf{x}) - \log p_\gamma(\hat{\epsilon}|\hat{\mathbf{z}}) \big)}_{\mathcal{L}_{\text{KLD}}}
\tag{51}
$$

# D ADDITIONAL EXPERIMENTS FOR MULTI-DOMAINS

## D.1 APPROACH

Figure 4 visualizes the data-generating process described in Eq. 5. Accordingly, the likelihood for this process, given the known auxiliary variable $\mathbf{u}$, is expressed as:

$$
p(\mathbf{z}, \epsilon, \mathbf{x}|\mathbf{u}) = p_\theta(\mathbf{x}|\mathbf{z}, \epsilon) p_\gamma(\epsilon|\mathbf{z}, \mathbf{u}) p_\delta(\mathbf{z}|\mathbf{u})
\tag{52}
$$

As a result, we redesign the encoder and prior module to learn the distribution $p_\gamma(\epsilon|\mathbf{z}, \mathbf{u})$ and $p_\delta(\mathbf{z}|\mathbf{u})$, as shown in Eq. 52, while keeping the decoder

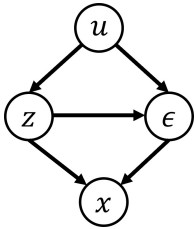

Figure 4: Visualization of the data generations of Eq. 5.

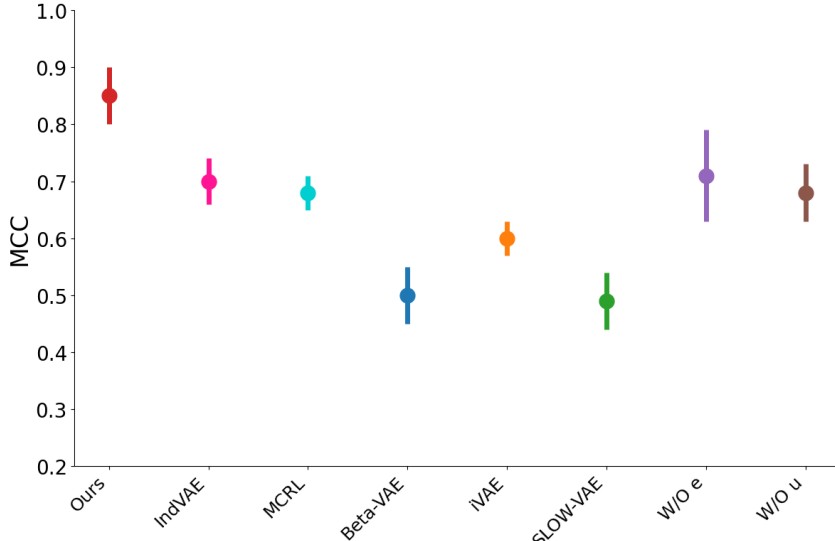

Figure 5: Mean Correlation Coefficient (MCC) scores from the multi-domain experiments comparing our framework with state-of-the-art approaches, including IndVAE, MCRL, BetaVAE, iVAE, and SlowVAE, as well as the baselines W/O $e$ and W/O $\mathbf{u}$.

unchanged. Accordingly, the ELBO is:

$$
\mathcal{L}_{\text{ELBO}} = \underbrace{\mathbb{E}_{\hat{\mathbf{z}} \sim q_\psi, \hat{\epsilon} \sim q_\phi} \left[ \log p_\theta(\hat{\mathbf{x}}^k | \hat{\mathbf{z}}, \hat{\epsilon}^k) \right]}_{\mathcal{L}_{\text{Recon}}}
$$
$$
\underbrace{-\beta_1 \mathbb{E}_{\hat{\mathbf{z}} \sim q_\psi} \left( \log q(\hat{\mathbf{z}} | \mathbf{x}) - \log p_\delta(\mathbf{z} | \mathbf{u}) \right) - \beta_2 \mathbb{E}_{\hat{\mathbf{z}} \sim q_\psi, \hat{\epsilon} \sim q_\phi} \left( \log q(\hat{\epsilon} | \mathbf{x}) - \log p_\gamma(\hat{\epsilon} | \hat{\mathbf{z}}, \mathbf{u}) \right)}_{\mathcal{L}_{\text{KLD}}}
$$
$$
(53)
$$

### D.2 ADDITIONAL EXPERIMENTS FOR MULTI-DOMAINS

**Synthetic Data Generation Process:** We adopt the data generation procedure from Kong et al. (2022); Li et al. (2023b) to synthesize data for multi-domain experiments. Specifically, we sample latent variables $\mathbf{z} \sim \mathcal{N}(\mu_u, \sigma_u^2 I)$, where domain-specific parameters $\mu_u \sim U(-4, 4)$ and $\sigma_u^2 \sim U(0.01, 1)$ are randomly drawn for each domain $u$. The remainder of the generation process aligns with Eq. 49, producing data from a total of five domains, i.e., $|\mathbf{u}| = 5$.

**Additional Experiments** We retain the implementation and training procedures described in Section C.1. Our approach is evaluated against IndVAE (Hu, 2008), which assumes conditional independence among observations given latent variables, as well as against MCRL (Sun et al., 2025), Beta-VAE (Higgins et al., 2016), iVAE (Khemakhem et al., 2020), and SLOW-VAE (Klindt et al., 2020). Furthermore, we conduct an ablation study involving the "W/O $e$" and "W/O u" baselines.

Figure 5 illustrates the MCC scores obtained in our multi-domain experiments. Our proposed method again achieves superior performance compared to all alternative approaches. This performance advantage can be traced back to our model's capability to disentangle latent variables $\mathbf{z}$ from dependent noise terms $\epsilon$, achieving the best identifiability under generalized dependency conditions. The comparative analysis with "W/O $e$" and "W/O u" highlights the impact of explicitly modeling $e$ and emphasizes the effectiveness of explicitly modeling $\mathbf{u}$.

### D.3 ADDITIONAL ABLATION STUDIES

**Ablations for higher dimensional $z$**

we additionally evaluated the scalability of our method to higher latent dimensions on the synthetic dataset by varying the latent dimensionality $N \in \{8, 12, 18\}$, while keeping the network architecture, training protocol, and all other hyperparameters fixed. Table 3 reports the MCC and compares with IndVAE Hu (2008): Even as the latent dimension increases, our method consistently achieves

| $N$ | IndVAE | Ours |
|---|---|---|
| 8 | $0.64 \pm 0.06$ | $0.80 \pm 0.02$ |
| 12 | $0.51 \pm 0.03$ | $0.68 \pm 0.05$ |
| 18 | $0.47 \pm 0.04$ | $0.61 \pm 0.05$ |

Table 3: MCC on the synthetic dataset for increasing latent dimensionality $N$.

higher MCC than IndVAE, indicating that the Jacobian-based sparsity regularization remains effective and that our approach scales well to higher-dimensional latent spaces within the considered regime.

**Ablations for hyperparameter sensitivity**

In our implementation on the synthetic dataset, we set the weight of the sparsity regularizer to $1$ and the KL weights to $\beta_1 = \beta_2 = 0.02$. In this section, we conducted a sensitivity analysis in which we vary *one* of these three hyperparameters at a time while keeping the others fixed at their default values. For all runs we use the same network architecture, batch size, number of epochs, learning rate, and training protocol as in the main experiments. The numbers reported below are MCC scores (mean $\pm$ std. over multiple runs) on the synthetic dataset. These results show the default setting $(\lambda, \beta_1, \beta_2) = (1, 0.02, 0.02)$ obtains the best results.

**Additional real-world experiments**

we expand our real-world evaluation beyond the original dataset and consider two additional person identity classification benchmarks. Specifically, we use the SYSU-MM01 dataset Wu et al. (2017) and the RobotPKU dataset Liu et al. (2017). We only use the RGB modality, since our focus is not on cross-modal person re-identification. SYSU-MM01 contains RGB images of $491$ identities from $6$ cameras, with a total of 30,071 images. RobotPKU contains more than 16,000 RGB images of $180$ identities, captured under dynamic robotic viewpoints. These datasets thus provide solid playgrounds for our experiments.

For performance comparison, we follow the same person index classification protocol as in our main experiment and compare against several state-of-the-art methods, including GTL (Yang et al., 2025), AGW (Ye et al., 2021), TransReID (He et al., 2021), CLIPReID (Li et al., 2023a), LDP-net Zhou et al. (2023), Style Fu et al. (2023), as well as MCRL (Sun et al., 2025) and IndVAE (Hu, 2008). Tables 5 and 6 report the Top-1 classification accuracy (mean $\pm$ std. over multiple runs).

To further examine the effect of the architecture choice, we replace the MLP in our framework on the MSTM-17 dataset with a single-layer Gated Recurrent Unit (GRU) Goodfellow et al. (2016) using the same hidden dimension (the classifier architecture and all training and evaluation protocols remain unchanged, and we set $N = M = 32$ for fair comparison). The resulting Top-1 accuracies are: GRU: $94.9 \pm 0.4$ versus our original MLP-based model: $94.4 \pm 0.7$. The GRU improves the classification results slightly against the MLP architecture. Overall, our method consistently outperforms strong baselines across three real-world person identity datasets.

# E  THE USE OF LARGE LANGUAGE MODELS (LLMS)

We use LLMs to detect and correct grammatical errors throughout the manuscript. No substantive edits requiring disclosure.

Table 4: Sensitivity of MCC to regularization hyperparameters.

| Hyperparameter | Value | MCC |
|---|---|---|
| $\lambda$ | 1 | $0.87 \pm 0.04$ |
| | 0.1 | $0.82 \pm 0.03$ |
| | 0.01 | $0.70 \pm 0.07$ |
| | 10 | $0.68 \pm 0.05$ |
| $\beta_1$ | 0.02 | $0.87 \pm 0.04$ |
| | 1 | $0.73 \pm 0.02$ |
| | 0.001 | $0.65 \pm 0.04$ |
| $\beta_2$ | 0.02 | $0.87 \pm 0.04$ |
| | 1 | $0.82 \pm 0.01$ |
| | 0.001 | $0.78 \pm 0.06$ |

Table 5: Comparison of Top-1 Accuracy on the ROBOTPKU dataset.

| Methods | Acc |
|---|---|
| AGW (Ye et al., 2021) | $87.6 \pm 0.8$ |
| TransReID (He et al., 2021) | $90.2 \pm 0.9$ |
| CLIPReID (Li et al., 2023a) | $91.7 \pm 1.1$ |
| GTL Yang et al. (2025) | $93.9 \pm 0.5$ |
| MCRL (Sun et al., 2025) | $94.5 \pm 1.0$ |
| IndVAE (Hu, 2008) | $95.8 \pm 0.8$ |
| Ours | $97.0 \pm 0.5$ |

Table 6: Comparison of Top-1 Accuracy on the SYSU-MM01 dataset.

| Methods | Acc |
|---|---|
| LDP-net Zhou et al. (2023) | $91.7 \pm 1.1$ |
| Style Fu et al. (2023) | $92.8 \pm 0.8$ |
| CLIPReID (Li et al., 2023a) | $94.1 \pm 1.0$ |
| GTL Yang et al. (2025) | $95.7 \pm 0.4$ |
| MCRL (Sun et al., 2025) | $96.4 \pm 0.8$ |
| IndVAE (Hu, 2008) | $96.8 \pm 0.5$ |
| Ours | $97.6 \pm 0.5$ |

