# OpenReview forum: "Provably Learning Representations under Generalized Dependency Structure"
_ICLR.cc/2026/Conference — Submitted to ICLR 2026_

### Official Review · Reviewer_TTFZ · 2025-10-21

**Soundness:** 2
**Presentation:** 2
**Contribution:** 1
**Rating:** 2
**Confidence:** 4

**Summary:**

This paper discusses the identifiability problem of latent variables with the existence of unknown dependent noise. Theoretical results are given that under some assumptions, latent variable z is subspace-identifiable and can be component-wise identifiable if sparsity condition is satisfied or auxiliary domain variable is further available. An identification algorithm is proposed by combining beta-VAE and a sparsity regularization term.

**Strengths:**

1. This paper focuses on latent variable identifiability, which is an important problem in representation learning.

2. This paper provides thorough results on this problem, including subspace/component-wise identifiability conditions and identification algorithm.

**Weaknesses:**

1.The problem setting of this paper is questionable. The existence of “generalized dependency structure” is the main challenge that this paper aims to address, i.e., the noise $\epsilon$ depends on latent variable $z$. However under other assumptions proposed by the author (g, e are injective, $\eta$ is independent), this problem is equivalent to one with independent noise: rewrite Eq. (1) as $x=g(z,e(z,\eta))\triangleq f(z,\eta)$, view $\eta$ as the noise (which is independent of $z$) instead of $\epsilon$. This observation undermines the main claim of this paper.

2.There is some clarity problem regarding Thm. 1. In assumption vi, I find no definition of the symbol $\tilde{h}$, so do the following symbols $\tilde{\mathcal{H}}$ and $\mathcal{T}$. This hinders me from further checking the proof. In addition, please give more explanation about why the uniqueness of $L_{x_b|z}$ implies the existence of $\tilde{h}$ as stated in Line 190-192, as well as in Line 820-822.

3.The assumptions in Thm 1. lack discussion. It is non-trivial to split $x$ into three parts $x_A, x_B, x_C$, what’s the difference among three parts or how do they function differently? Can the assumptions in Thm 1. be easily satisfied? Did you check whether they are satisfied in your synthetic experiments? Since Thm 1. is the core of this work, such discussions should be included.

4.The novelty of component-wise identifiability results (Thm. 2/3) is limited. In my point of view, these 2 theorems assume the conditions in Thm. 1 are satisfied, i.e., latent variable z is already subspace-identified. In this case, the component-wise identifiability problem of z becomes exactly the setting in the work of Kong et al. (2022) and Zhang et al. (2024), therefore the provided results are direct corollary of previous theorems. The introduced dependency structure does not bring novel challenge to the proof. Please correct me if there are misunderstandings.

**Questions:**

See weaknesses for main questions. Here are some minor questions:

1.In Def. 2, what is a “permutation”? Is it an index permutation on the dimensions of z? If this is the case, what does $\pi(z^n)$ mean? $z^n$ has only one dimension.

2.Since Thm.1 originates from the work of Hu (2008), can you provide some intuition about the main difficulty caused by the noise dependency for the proof, and your key idea for the mitigation?

---

> ### Author Response · Authors · 2025-11-21
> **Rebuttal by Authors**
>
> >Q6: Since Thm.1 originates from the work of Hu (2008), can you provide some intuition about the main difficulty caused by the noise dependency for the proof, and your key idea for the mitigation
>
> A: We believe that prioritizing Q6 would be very helpful for understanding the contribution of our work.
>
> To the best of our understanding, the main difficulty introduced by *dependent noise* is to establish an invertible, smooth transformation $h$ between the learned latent variables $\hat{z}$ and the true latent variables $z$, i.e., to show that $\hat{z} = h(z)$ still holds when the noise term $\epsilon$ depends on $z$. Below we contrast the classical cases with our setting and highlight our key idea.
>
> (1) Noise-free setting:
>
> When the data-generating process is $x = g(z)$ with $g$ injective and smooth, observational equivalence (Def.~1) implies
> $x = g(z) = \hat{g}(\hat{z})$. Injectivity of $g$ (and $\hat{g}$) yields
>
> $\hat{z} = \hat{g}^{-1} \circ g (z) =: h(z)$
>
> so $h$ exists and is invertible. This is the standard argument used in [4,23].
>
> (2) Independent-noise setting:
>
> With the noise term $\epsilon$ that is independent of $z$, the data-generating process is $x = g(z,\epsilon)$ with $p(\epsilon | z) = p(\epsilon)$, where $g$ still is injective and smooth. The standard solution is to first show that there exists an invertible map
> $$
> \bar{h} : (z,\epsilon) \rightarrow (\hat{z},\hat{\epsilon})
> $$
> relating any two observationally equivalent models. One then might need to impose additional conditions, such as the additive independent noise in [10], or the linear independence of the columns of the Jacobian of $g$ with respect to $z$ in MCRL [5], to prove that $\frac{\partial \hat{z}}{\partial \epsilon}=0$. In other words, we have to make sure $\hat{z}$ and $\epsilon$ is separable.
>
> (3) Dependent-noise setting:
>
> if $\exists\bar{h} : (z,\epsilon) \rightarrow (\hat{z},\hat{\epsilon})$ (where $\bar{h}$ denotes an invertible transformation),
> when the noise term $\epsilon$ and $z$ is statistically dependent ($p(\epsilon | z) \neq p(\epsilon)$), we can **NOT** enforce $\frac{\partial \hat{z}}{\partial \epsilon}=0$ (if $\bar{h}$ is differentiable) by a standard strategy such as deconvolution in [10] for handling the independent noise, since $\epsilon$ carries information about $z$.
>
> Addressing the aforementioned challenge, the previous works, such as [1], impose conditional independence assumptions on partitions of $x$ given $z$ (see details in Sec.3 and Sec.~B.1 in our paper).
>
> (4) Our key idea:
>
> However, the conditional independence assumption might not always hold in practice. We provide an example to illustrate this from L.36 to L.42 in the paper. Therefore, our key idea explicitly models the generalized dependency among partitions of $x$ given $z$ through Eq. (4) in the paper, and then analyze it through the operator perturbation theory. Under Assumptions $i\sim vi$, Theorem1 yields subspace identifiability, i.e., $\hat{z} = h(z)$, even in the presence of noise that depends on $z$ (please see details in the proof of Theorem 1 in Sec.B.2.).

---

> ### Author Response · Authors · 2025-11-21
> **Rebuttal by Authors**
>
> >Q1: The problem setting of this paper.
>
> A: 1. Clarification on the definition of "generalized dependency" and our key setup
>
> We would like to first clarify that our notion of “generalized dependency” does **NOT** mean “the noise $\epsilon$ depends on the latent variable $z$.” In both original paper and paper revision, as defined at the end of the first paragraph in Sec. 1, in the last paragraph of Sec. 3, and in Assumption $i$ of Thm. 1, etc., the **generalized dependency structure refers to allowing arbitrary dependencies among observed variables conditional on $z$**. In particular, we do not assume the conditional independence that proposed in [1] to handle dependent noise: $p(x|z) = p(x_a|z)p(x_b|z)p(x_c|z)$, and instead allow $p(x|z) \neq p(x_a|z)p(x_b|z)p(x_c|z)$. Our key setup is to establish identifiability of $z$ under this generalized conditional dependency structure when the dependent noise presents, **NOT** to handle dependent noise only. To the best of our knowledge, this is the first identifiability result under generalized dependency structure with dependent noise.
>
> 2. Clarification on "dependent noise"
>
> Based on our discussions in answer to Q6, we emphasize that $x=g(z,\epsilon)=g(z,e(z,\eta))\triangleq f(z,\eta)$ does **NOT** justify treating our model as equivalent to a standard independent noise condition common in the literature. This is because that:
>
> (1) By definition, $f(z,\eta) \triangleq g(z,e(z,\eta)) = g(z,\epsilon) = x$ suggests **the dependent noise $\epsilon$ implicitly presents inside $f$**. In other words, this definition does **NOT** remove the dependent noise that our setup relies on.
>
> To illustrate this point, since both $g$ and $e$ are smooth functions, taking derivative w.r.t. $z$ on both sides of $f(z,\eta) = g(z,e(z,\eta))$ results in:
> $$\frac{\partial x}{\partial z}  = \frac{\partial f(z,\eta)}{\partial z} = \frac{\partial g(z,\epsilon)}{\partial z} + \frac{\partial \epsilon}{\partial z}$$
> Notably, $\frac{\partial \epsilon}{\partial z}\neq 0$ given $\epsilon=e(z,\eta)$. This suggests $x = f(z,\eta)$ does not convert the model into one with independent noise setting as we discussed previously; the dependent noise $\epsilon = e(z,\eta)$ remains inside $f$.
>
> For instance, let $z\in\mathbb{R}$ be the latent variable, $\eta\in\mathbb{R}$ denotes an exogenous variable independent of $z$, $\epsilon=(\epsilon_1,\epsilon_2)^\top\in\mathbb{R}^2$ the dependent noise, and $x\in\mathbb{R}^3$ the observation. Let us define the data generating process $x=g(z,\epsilon),\quad \epsilon=e(z,\eta)$ in the following:
> $$
> \epsilon = e(z,\eta) \triangleq
> (
> z^3,
> \eta^3 + z)^\top
> \quad
> x=g(z,\epsilon) \triangleq
> (
> z,
> \epsilon_1,
> \tanh(\epsilon_2)^\top
> $$
>
> Now define
> \begin{align}
> f(z,\eta) \triangleq g\bigl(z,e(z,\eta)\bigr)=
> (
> z,
> z^3,
> \tanh(\eta^3 + z)
> )^\top
> \end{align}
>
> Although the model is now written as $x = f(z,\eta)$, the last two rows $(z^3,\eta^3 + z)^\top$ still depend on the definition of $e(z,\eta)$.
> **Therefore, to establish an invertible transformation between $\hat{z}$ and $z$ over $x=f(z,\eta)$ still requires to prove that $\hat{z}$ is not a function of $e(z,\eta)$.**
>
> (2) We do not impose any of the structural assumptions on the exogenous variable $\eta$ (or on $f$) that are required to disentangle $z$ and $\eta$ in independent–noise models, such as additive noise in iVAE [10], or linear independence of the jacobians of $f$ in MCRL [5].
>
> (3) Additionally, if one discards $\epsilon = e(z,\eta)$ and treats $f$ as an unconstrained conditional, the class of $f$ is strictly larger: since $\eta$ is exogenous, any invertible transformation of $z$ can meet the observational equivalence as formalize in Definition~1. With such an unconstrained $f$, $z$ is not identifiable without further assumptions.
>
> *We welcome further discussion on this point, if the reviewer can specify which assumption in our work can work on $f$ would justify dropping the dependent noise term $\epsilon$ while still obtaining $\hat{z}=h(z)$.*

---

> ### Author Response · Authors · 2025-11-21
> **Rebuttal by Authors**
>
> >Q2: clarity problem regarding Thm. 1
>
> A: 1. The definition of $\tilde{h},\tilde{\mathcal{H}},\mathcal{T}$
>
> We have defined the $\tilde{h}: \mathcal{Z}\rightarrow\mathcal{Z}$ as an invertible transformation, where $\mathcal{Z}$ denotes the space of the latent varaible $z$. $\tilde{\mathcal{H}}$ denotes the function class of $\tilde{h}$ and $\mathcal{T}$ denotes the function class of $t$. These notions are defined from L.179 to L.181, as well as L.191 to L.193 in the original paper.
>
> In light of your suggestion, we have added the following description to Assumption $vi$ in Thm.1 to improve the clarity:
> "Assumption $i\sim v$ results in the existence of an invertible transformation $\tilde{h}:\mathcal{Z}\rightarrow\mathcal{Z}$, such that $\tilde{z}=\tilde{h}(z)$."
>
> 2. The existence of invertible transformation $\tilde{h}$
>
> In our setting, $L_{x_b|z}$ play the role of the "eigenvalues" of the decomposition of $L_{x_a|z} L_{x_b|z} L_{x_a|z}^{-1}$.
> The uniqueness of the spectrum of $L_{x_b|z}$ means that it is determined by such decomposition. Specifically, permute $z$ introduces a permute operator $P$, such that, $P L_{x_b|z} P^{-1} = L_{x_b|\bar{h}(z)}$, where $\bar{h}$ denotes a permute transformation.
> Therefore, associated eigenfunctions $L_{x_a|z}$ and $L_{x_a|z}^{-1}$ are permuted, but the eigenvalues $L_{x_b|z}$ remains invariant. Notably, we use the term "eigenvalue" by following [1]. Since $\bar{h}$ is a permute transformation, it has to be invertible. Therefore, we can reach to our conclusion.
>
> In light of your suggestion, we have added the above discussion to L.874 to L.878 in the paper revision.
>
> >Q3: Discussions of assumptions in Thm 1 (P1)
>
> A: 1. The impact of $x_a,x_b,x_c$:
>
> By ``function differently'', we assume you were asking the impacts of $x_a,x_b,x_c$ w.r.t. the subspace identifiability results from Thm.1.
>
> Our goal in Thm.~1 is to prove that any observationally equivalent model must satisfy $\hat{z} = h(z)$. For this identifiability statement, the particular labeling of the three partitions of $x$ is irrelevant. Specifically, we do **NOT** assume that the three parts have their particular individual roles in the data. The only difference between $x_a, x_b, x_c$ in the theorem is how they enter the operators used in the proof (e.g., they are used in $L_{x_a, x_b |x_c} L_{x_a |x_c}^{-1}$). Any permutation of the three partitions yields an equivalent set of assumptions and the same identifiability conclusion.
>
> In light of your suggestions, we have incorporated the following discussion into the first paragraph of Sec.4 in the paper:"The partition $\mathbf{x} = (x_a, x_b, x_c)$ is arbitrary; Our analysis only requires that there exist these partitions, and is invariant to any relabeling of these partitions."
>
> 2. Discussion of the assumptions in Thm.1:
>
> Unlike prior works [1] that assume conditional independence of observed variables given $z$, we relax this in Assumption $i$ by allowing $p(x | z) \neq p(x_a | z) , p(x_b | z) , p(x_c | z)$, where $x = (x_a, x_b, x_c)$. A concrete example of such dependent measurements is given from L.35 to L.42 of the introduction. Assumption $i$ captures this generalized dependency structure by allowing such dependencies to exist.
>
> Assumption~$ii$ requires the conditional operators $L_{x_a|z}$, $L_{x_b|z}$, and $L_{x_c|z}$ to be injective.  This is a mild *functional completeness* condition in nonparametric factor analysis like [1,2]. Intuitively, it ensures that each block $x_a, x_b, x_c$ exhibits sufficient variation as $z$ changes, so that functions of $z$ can be recovered from the corresponding conditional distributions.
>
> Assumptions $iii\sim iv$ are spectral regularity conditions on the operator $L_{x_a,x_b|x_c}L_{x_a|x_c}^{-1}$. Assumption $iii$ requires this operator to have distinct eigenvalues, and Assumption $iv$ assumes a self-adjoint structure so that we can leverage the Weyl's inequality. Together with the spectral–gap and perturbation bound in Assumption~($v$), which defines $\overline{Per} < \kappa$ (where $\kappa := \min_{i \neq j} \frac{|\rho^i - \rho^j| - \alpha}{2} \geq 0$), ensure that the separation between any pair of unperturbed eigenvalues exceeds $2 \times\overline{Per}$. Consequently, the discs remain disjoint: $D^i \cap D^j = \emptyset \quad \forall i \neq j$. This guarantees that the $\rho^i_\Lambda$ remains to be distinct from $\rho^j_\Lambda$.
>
> Also, Assumption $vi$ assumes a smooth transformation $t\in T: \mathcal{Z} \to \mathcal{Z}$. This smoothness and invertibility constraint is standard in nonlinear identifiability results [3-17]. Theorem 1 thus concludes that $\exists h\in H' \cap T$ such that $\hat{z}=h(z)$ even under generalized dependency structure. Thus, we can disentangle $z$ from the noise term $\epsilon$.

---

> ### Author Response · Authors · 2025-11-21
> **Rebuttal by Authors**
>
> >Q3: Discussions of assumptions in Thm 1 (P2)
>
> 3. The satisfaction of the assumptions in Thm.1 in sythetic experiment:
>
> Given the definition 4 in the paper, all operator assumptions in Theorem1 can be explained in terms of the conditional densities $p(x_a| z)$, $p(x_b| z)$, $p(x_c| z)$, etc., induced by our data synthesizing process.
>
> We summarize the data synthesizing process in Eq.49 in Sec.C.1 in our paper in the following:
> $$
>     \epsilon_1=\mathcal{N}(0, (z_1)^2), \quad \tilde{g}_1(z_1,\epsilon_1) = sinh(z_1)\times \epsilon_1
> $$
> $$
>     \epsilon_2=tanh(z_2), \quad \tilde{g}_2(z_2,\epsilon_2) = \frac{1}{1+exp(z_2)}\times \epsilon_2
> $$
> $$
>     \epsilon_3=Laplace(0,|z_3|), \quad  \tilde{g}_3(z_3,\epsilon_3) = (z_3)^2\times \epsilon_3
> $$
> $$
>     \tilde{x}_m = \tilde{g}_m(z_m,\epsilon_m), \quad x = g(\tilde{x}) = \sigma (R \tilde{x})
> $$
> where $m=\{1,2,3\}$, $z_m\sim U(0,1)$, $R\in\mathbb R^{3\times 3}$ is a fixed full-rank matrix with sparse, small nonzero off-diagonal entries. $\tilde x=(\tilde x_1,\tilde x_2,\tilde x_3)^\top$, and $\sigma$ is a smooth strictly monotone scalar nonlinearity applied coordinate-wise.
>
> (1) Assumption $i$:
>
> Conditional on $z$, $\tilde x_m$ is independent of each other. Multiplication by $R$ yields $p(x| z)\neq p(x_a| z)\,p(x_b| z)\,p(x_c| z)$ for all $z$ in the support, given the existence of the nonzero off-diagomal entries in $R$. This directly enforces Assumption $i$.
>
> (2) Assumption $ii$:
>
> In our simulator, each $\tilde x_m$ is generated from $z_m$ via either a non–degenerate Gaussian/Laplace location–scale family with strictly positive, smooth scale depending on $z_m$ or via a smooth strictly monotone deterministic map of $z_m$. Thus $z_m\neq z_{m'}$ implies conditional laws, i.e., different operators $L(\tilde x_m|z_m)\neq L(\tilde x_m|z_{m'})$ for $\tilde{x}_m$.
>
> $R$ has full rank and we set it as a small perturbation matrix from the diagonal one,
> and $\sigma$ is strictly monotone. Thus, any change in $z_m$ would eventually lead to a change in the conditional distribution of $x_a$ and $x_c$. In other words, $L_{x_a|z}, L_{z|x_c}$ are injective. Finally, $L_{x_a|x_c}=L_{x_a|z}L_{z|x_c}$ is a composition of two injective operators and hence injective. Therefore Assumption $ii$ holds by construction in our synthetic model.
>
> (3) Assumption $iii$:
>
> In our generator there are exactly three independent latent sources $z_1,z_2,z_3$, and $x_b$ depends smoothly and non-trivially on all of them through $R$ and $\sigma$. Hence the family $\{p(x_b|z)\}$ is genuinely 3–dimensional, and $L_{x_b|z}$ has three non-zero eigenvalues.
> The only variation between $x_a$ and $x_b$ also comes from the three-dimensional $z$. Consequently $L_{x_a,x_b|x_c}L_{x_a|x_c}^{-1}$ has exactly the a three-dimensional non-zero eigenvalues as well.
> Since $R$ is full rank with small nonzero off-diagonal entries, our $R$ enables these three eigenvalues are all distinct and well separated. Thus Assumptionm $iii$ is satisfied by construction.
>
> (4) Assumption $iv$:
>
> In our synthetic construction we enforce this directly through the mixing matrix $R$. We first fix an invertible row block $R_a\in\mathbb R^{3\times 3}$ corresponding to $x_a$, then choose any symmetric matrix $B\in\mathbb R^{3\times 3}$ with distinct real eigenvalues, such as $B=\mathrm{diag}(1,2,3)$, and set $R_b := B R_a$, for the row associated with the $x_b$. On the space of $z$, the transform $L_{x_a|z}L_{x_b|z}L_{x_a|z}^{-1}$ is then represented by the matrix $B = R_b R_a^{-1}$, which is symmetric by construction and hence defines a self-adjoint operator.
> Thus Assumption $iv$ is satisfied by construction in our synthetic generator.
>
> (5) Assumption $v$:
>
> In the synthetic experiment we keep $R$ with only a few off-diagonal entries. Since all the operators involved depend on the entries of $R$, the upper bound of the perturbatoin operator $\overline{Per}$ remains strictly smaller than the eigen-gap for $R$ in a sufficiently small neighbourhood of the diagonal choice. Iff $R$ in that regime, the perturbation condition in Assumption $v$ holds by construction.
>
> (6) Assumption $vi$:
>
> Let us consider the row of $R$ defining $x_b$, and assume that $x_b$ depends monotonically on $\tilde x_2$, and then apply the strictly monotone nonlinearity $\sigma$. Thus $x_b$ remains a smooth, strictly monotone function of $z$ given the smoothness and strictly monotone of $\tilde{g}_2$.
>
> Therefore, we can easily define that
> $$
> t(z) := \mathbb E[x_b|Z=z] = \int x_b\,p(x_b|z)\,dx_b = M(L_{x_b|z})
> $$
> where $t(z)$ is a differentiable and strictly monotone function of $z$. Hence Assumption $vi$ holds by construction.

---

> ### Author Response · Authors · 2025-11-21
> **Rebuttal by Authors**
>
> >Q4: The novelty of Thm.2 and Thm.3:
>
> A: The main contribution of this work focuses on proving the identifiability under the generalized dependency structure.
> Once the subspace identifiability results for $z$ established, we are able to leverage the Theorem 2 and 3 to further prove the component-wise identifiability.
>
> Our main theoretical contribution is Theorem 1, which establishes *subspace identifiability* of $z$ under the generalized dependent–noise setting $\epsilon = e(z,\eta)$ and arbitrary dependencies among observed variables. This setting is not covered by [1,4,23], all of which either assume independent noise or conditional independent observations, and therefore cannot be directly applied when $\epsilon$ depends on $z$. Once Theorem1 guarantees that the subspace identifiability, we can leverage Theorem 2 and 3 to obtain component-wise identifiability of $z$.
>
> In light of your suggestions, we have renamed Theorem 2 and 3 to "Corollary 2'' and "Corollary 3'', respectively.
>
> >Q5: The definition of permutation
>
> A: Thanks for your question. By $\pi(z^n)$, we would like to express the $n-$dimension of $z$ after being through the permutated transformation $\pi$. In light of your suggestion, we have changed the phrase to $z^{\pi(n)}$.
>
> --*Reference*
>
> 1 Hu. Instrumental variable treatment of nonclassical measurement error models. Econometrica, 2008.
>
> 2 Mattner. Some incomplete but boundedly complete location families. The Annals of Statistics, 1993.
>
> 3 Liang, et al. Causal component analysis. NeurIPS 2023
>
> 4 Kong, et al. Partial disentanglement for domain adaptation. ICML 2022
>
> 5 Sun,et al. Causal representation learning from multimodal biomedical observations. ICLR 2025
>
> 6 Kong, et al. Identification of nonlinear latent hierarchical models. NeurIPS 2023
>
> 7 Zheng, et al. Nonparametric factor analysis1 and beyond. AISTATS 2025
>
> 8 Khemakhem, et al. Variational autoencoders and nonlinear ica: A unifying framework. AISTAS, 2020
>
> 9 Morioka et al. Causal representation learning made identifiable by grouping of observational variables. ICML 2024
>
> 10 Li, et al. Subspace identification for multi-source domain adaptation. NeurIPS 2023.
>
> 11 Chen, et al. Caring: Learning temporal causal representation under non-invertible generation process. ICML 2024
>
> 12 Lachapelle, et al. Synergies between disentanglement and sparsity: Generalization and identifiability in multi-task learning. ICML, 2023
>
> 13 Zheng, et al. Generalizing nonlinear ica beyond structural sparsity. NeurIPS 2023
>
> 14 Li, et al. Learning causal domain-invariant temporal dynamics for few-shot action recognition. ICML 2024
>
> 15 Li, et al. Identification of intermittent temporal latent process. ICLR 2025
>
> 16 Ng, et al. A general representation-based approach to multi-source domain adaptation. ICML 2025
>
> 17 Zheng, et al. On the identifiability of nonlinear ica: Sparsity and beyond. NeurIPS, 2022

---

> ### Comment · Reviewer_TTFZ · 2025-11-25
> **Discussion on Q1**
>
> Thanks for the rebuttal. Since Q1 is my primary concern, let's focus on this part first.
>
> After your explanation on "generalized dependency", now I understand it refers to "dependency among observed parts conditioned on z" rather than "$\epsilon$ depends on z". Please revise the last sentence of your first paragraph in Sec. 1 by canceling the italics of text "when the dependent noise presents", or it would cause confusion.
>
> Even though, my concern regarding the problem definition remains, since this work still builds on the challenge of $\epsilon$-$z$ dependence. I understand that after reformulation using f, the dependence between $\epsilon$ and $z$ remains. However, now $\eta$ is independent from $z$ now, and there is no way to distinguish these two formulations (without further assumptions). In your reply to Q6 part (2), you have stated that there are existing approaches to handle problems with independent noise, so why not view this problem as identifying $z$ with independent noise $\eta$? If your reason is that you do not want to impose those assumption introduced in [10] or [5], please discuss why your assumption is a better choice in practice (e.g., why it is easier to be satisfied).
>
> I notice that you have discussed that "the class of $f$ is strictly larger", but I don't understand why you view $f$ as "unconstrained" but $g$ as "constrained". Are there any obstacles to replace $\epsilon$ with $\eta$ and $g$ with $f$？If this is possible without affecting your Thm. 1, then you do not need to propose "dependent noise" as one of your challenge in this paper.

---

> ### Author Response · Authors · 2025-11-25
> **Follow-up rebuttal on Q1**
>
> First of all, we would like to clarify that our key contribution is to idenfity $z$ under the generalized dependency structure. **Your primary concern of our work regarding "$\epsilon$–$z$ dependence", however, is *NOT* the focus of our work**.
>
> Notably, we introduce the dependent noise term $\epsilon$ in Eq.(1) to remain consistent with the standard notion in [3,4,5], which introduce "dependent" noise term and study identifiability under conditional independence assumptions. *Using $\epsilon$ makes it clear that our data-generating process lies in the same dependent-noise family as these works, while allowing us to relax their conditional-independence constraint*. As discussed in the first paragraph of the introduction and in Sec. 3, the conditional-independence constraint imposed in [3,4,5] directly motivates our generalized dependency structure.
>
> Apart from the concern that dropping $\epsilon$ would obscure the consistency between our work and previous ones, we are **NOT** opposed to rewriting $x = g(z,\epsilon), \epsilon = e(z,\eta)$ as $x = f(z,\eta)$.
> What we emphasized in Point 2 of our answer to Q1, and in Point 2,3 of our answer to Q6 are, this notational reformulation does **NOT** mean that our problem reduces to identifying $z$ under a standard “independent noise” setup as studied in the existing literature [1,2]. *Those works impose additional structural assumptions that we do **NOT** assume*. We recapitulate and summarize our previous response to Q6 and Q1 in the following:
>
> - 1. **Additive independent noise vs. general latent noise**
>
> IVAE [1] assumes an additive, independent noise structure in their Eq. (6) for the data generating process. In contrast, our formulation of $\epsilon$ relaxes the independent requirement, and we do not impose the additivity. Thus, our setup is more generalized than the additive models considered in iVAE.
>
> - 2. **Multimodal vs. single-modal observations**.
>
> In additional to the difference between the independent and our general latent noise, MCRL [2] assumes access to multimodal observations and uses linear independence of the Jacobians of their mixing functions w.r.t. $z$ across modalities to disentangle $z$ from $\eta$. Their subspace identifiability result relies on having multiple modalities. When only a single modality is available, the guarantees in [2] no longer hold. In contrast, our Theorem 1 establishes the subspace identifiability of $z$ even when there is only one modality.
>
> Finally, by “unconstrained $f$” we mean that, if we express the data-generating process as $x = f(z,\eta)$ without any assumptions on $f$, $z$ will not be identified.
>
> In light of your suggestion, we have removed the phrase “when the dependent noise presents” from the introduction to avoid confusion. **We would also appreciate if you can clarify on whether, and which specific assumptions in the literature with independent noise setup, one can obtain subspace identifiability of $z$ under generalized dependency structure, i.e., why $\hat{z}$ is not a function of $e(z,\eta)$ that exists inside $f$**.
>
> --*References*
>
> 1 Khemakhem, et al. Variational autoencoders and nonlinear ica: A unifying framework. AISTAS, 2020
>
> 2 Sun, et al. Causal representation learning from multimodal biomedical observations. ICLR 2025
>
> 3 Hu. Instrumental variable treatment of nonclassical measurement error models. Econometrica, 2008.
>
> 4 Fu, et al. Identification of nonparametric dynamic causal model and latent process for climate analysis. Arxiv, 2025
>
> 5 Zheng, et al. Nonparametric Factor Analysis and Beyond. AISTATS 2025

---

### Official Review · Reviewer_Dt9i · 2025-10-29

**Soundness:** 3
**Presentation:** 3
**Contribution:** 3
**Rating:** 4
**Confidence:** 3

**Summary:**

The paper studies the identifiability when observations remain dependent conditioning on the latent factors. It proposes a generalized dependency framework with theoretical guarantees for both subspace and component-wise identifiability, supported by perturbation-based analysis and sparsity assumptions. A corresponding variational inference model is developed and shown to outperform prior methods on synthetic and real-world datasets, demonstrating promising theoretical and empirical results.

**Strengths:**

In my view, the main contribution of this paper lies in its theoretical advancement, which allows dependencies among observations given latent variables with novel assumptions. The experimental results on the synthetic dataset are also promising. The paper is clearly written, and I appreciate the well-organized presentation and thorough background provided by the authors.

**Weaknesses:**

I think one shortcoming of the paper is the lack of discussion of Li et al. (2025) [https://www.arxiv.org/pdf/2510.18281]. Specifically, Li et al. also build their identifiability results upon Hu (2008) and consider a generalized dependency structure among observations. Since Li’s work was released earlier, it would be important to include a detailed discussion clarifying the differences and the unique contributions of the current paper relative to theirs.

Another notable concern is the mismatch between the reported baseline accuracies and those reported in the original papers for the real-world dataset. I cross-checked the results of AGW (Ye et al., 2021), TransReID (He et al., 2021), and CLIP-ReID (Li et al., 2023a). The Rank-1 (R1) accuracies reported in the original works are considerably lower than those presented in this submission — for example, the AGW paper reports an R1 of 68.3, whereas this paper lists 85.5. I recommend the authors clarify how these baseline results were obtained, ensuring that all comparisons are fair and consistent with the original implementations (an anonymous link containing the code implementation will be appreciated).

**Questions:**

Please see the weakness

---

> ### Author Response · Authors · 2025-11-21
> **Rebuttal by Authors**
>
> >Q1: Discussion of Li et al. (2025) [https://www.arxiv.org/pdf/2510.18281]
>
> A: Thanks for your question.
> We would like to first clarify that [1] does **NOT** introduce generalized dependency structure. We would like to address our settings that are different from [1], and then explain more precisely what we mean by ``generalized dependency''.
>
> 1. **Dependent noise vs. independent noise**:
>
> In our model, the noise term $\epsilon$ is allowed to depend on the latent variable $z$; specifically, we consider $\epsilon = e(z,\eta)$ in Eq. (1) of our paper. In contrast, the data-generating process in Eq. (1) of [1] assumes that the noise is independent of $z$. Therefore, [1] can be viewed as a special case of our framework obtained by restricting $\epsilon$ to be independent of $z$.
>
> 2. **Generalized dependency structure vs. conditional independence**:
>
> Assumption $i$ in our Theorem1 allows $(x_a,x_b,x_c)$ to exhibit arbitrary dependence conditioning on $z$. We do not impose any conditional independence constraints among these observed variables. By contrast, [1] imposes conditional independence given their Markov structure among the "4 Adjacent Observed Variables'' as formalized in Eqs. (A12) and (A15) of [1]. As a consequence, the identifiability result of [1] does not apply when these conditional independence relations are violated, whereas our result still applies to such settings with fully general dependence among the observations.
>
> Given these two essential differences, our identifiability result for recovering $z$ strictly generalizes the one in [1].
>
> In light of your suggestion, we clarify the definition of *generalized dependency* in the introduction from L.54 to L.56 in the paper revision as follows:"In this paper, we use the term generalized dependency to refer to data-generating processes in which the observed variables are allowed to have arbitrary dependence conditioning on the latent variable, without the conditional independence restrictions imposed in [1]."
>
> Additionally, [1] went public in October, which is after the due of ICLR 2026. **We welcome further discussions if the reviewer could specify where the generalized dependency structure in our Assumption $i$ of Theorem 1, in your view, was already explicitly introduced in [1].**

---

> ### Author Response · Authors · 2025-11-21
> **Rebuttal by Authors**
>
> > Q2: Clarifications regarding our reported results
>
> A: The mismatch between our reported numbers and those in the original papers is due to $1.$ a different evaluation *metric*; $2.$ a different experimental *setup*.
>
> **1. Metric: Top-1 classification vs.\ Rank-1 ReID:**
>
> As described from L.430 to L.431 in the original paper, in our real-world experiment we cast the problem as an Person Index classification task. We first train a VAE using our objective in Eq.~(12), obtain the latent representation $z$, and then train a classifier to predict the person index $y$ from $z$, i.e., to model $p(y |z)$. We highlight that we report Top-1 classification accuracy for the classifcation in the caption of Table 1 in the paper.
>
> In contrast, AGW, TransReID and CLIP-ReID are originally evaluated on MSMT17 as person ReID methods, where the standard metric is Rank-1: given a query image, one computes feature similarities to a gallery and measures whether the correct identity is ranked first. Top-1 classification accuracy and Rank-1 ReID accuracy are therefore not directly comparable.
>
> We use their public code to train on MSMT17 dataset to obtain the appearance feature, and then report the Top-1 classification accuracy under our classification protocol, not Rank-1 score.
>
> **2. Setup: standard classification vs. ReID:**
>
> For fairness, the comparing methods in Table~1, including AGW, TransReID, CLIP-ReID, GTL are trained under the same supervised classification protocol on MSMT17: We use 60\% of the data for training, 20\% for validation and the rest 20\% for test. We have stated this dataset splitting protocal from L.449 to L. 456 in the paper. In particular, we use their public code to first train on the training dataset to obtain the features, and then append a classifier to obtain the final classification score. Notably, we ask for the code from the authors of [2] to conduct our experiments instead of their few-shot settings. The difference in supervision (full-data classification vs.\ few-shot ReID) also contributes to the higher accuracies we report compared to the original works.
>
> **3. Why we use classification instead of ReID:**
>
> Classification has been widely adopted to empirically assess identifiability like [3-11]. Following this line of work, we use a closed-set classification task to support identifiability results in our work: after learning the latent representation $z$, we train a classifier to predict the identity label $y$ from $z$. High classification accuracy then directly reflects how well $z$ preserves the information relevant for $y$.
>
> In contrast, standard person ReID benchmarks focus on an setting, where the training classes are “not available” and the testing classes (queries) are denoted as previously “unseen” [12]. This protocol introduces additional factors—such as metric learning, similarity computation, and nearest-neighbor search—which add approximation and estimation error that are orthogonal to our goal. To cleanly isolate and evaluate identifiability of the latent variables, we therefore opt for the simpler and more direct classification setup rather than the conventional ReID protocol.
>
> Our implementation is built on an internal codebase that is currently being used for another ongoing project. We will release the code upon the progress of the current project.
>
> --*Reference*
>
> 1 Li, et al. Online time series forecasting with theoretical guarantees. NeurIPS 2025
>
> 2 Yang, et al. Cross-modal few-shot learning: a generative transfer learning framework, 2025. Arxiv 2024
>
> 3 Kong, et al. Partial disentanglement for domain adaptation. ICML 2022
>
> 4 Sun,et al. Causal representation learning from multimodal biomedical observations. ICLR 2025
>
> 5 Khemakhem, et al. Variational autoencoders and nonlinear ica: A unifying framework. AISTAS, 2020
>
> 6 Chen, et al. Caring: Learning temporal causal representation under non-invertible generation process. ICML 2024
>
> 7 Lachapelle, et al. Synergies between disentanglement and sparsity: Generalization and identifiability in multi-task learning. ICML, 2023
>
> 8 Zheng, et al. Generalizing nonlinear ica beyond structural sparsity. NeurIPS 2023
>
> 9 Li, et al. Learning causal domain-invariant temporal dynamics for few-shot action recognition. ICML 2024
>
> 10 Li, et al. Identification of intermittent temporal latent process. ICLR 2025
>
> 11 Ng, et al. A general representation-based approach to multi-source domain adaptation. ICML 2025
>
> 12 Zheng, Liang, Yi Yang, and Alexander G. Hauptmann. "Person re-identification: Past, present and future." Arxiv 2016

---

### Official Review · Reviewer_Mn5G · 2025-10-30

**Soundness:** 3
**Presentation:** 2
**Contribution:** 3
**Rating:** 6
**Confidence:** 2

**Summary:**

The authors study the identifiability of latent variables under the condition that the noise is dependent on the latent variables. Under some assumptions, the authors prove the subspace and component-wise identifiability. These results motivate the design of the VAE structure and the learning objective. Using their model, the authors show that the proposed method gets better identifiability on the synthetic data, and better accuracy on a downstreaming classification task on a real data set, which indicates better latent variables.

**Strengths:**

1. The work extends the previous setting of independent noise to dependent noise. This is a practical setting, and renders more flexibility to the model.

2. Based on the theoretical analysis, there are several improvements to traditional VAE, such as the conditional noise distribution and sparsity regularization. Some ablations show the effectiveness of these improvements.

**Weaknesses:**

1. The authors add the regularization on the Jacobian to meet the sparsity assumption. This may be inefficient for high dimensional data.

2. There are three regularization terms, hence three hyperparameters for their weights. There is no sensitivity analysis of them.

3. For real data analysis, there is only one data set. Furthermore, it consists of images. I am afraid the MLP architecture may not be optimal for images.

**Questions:**

Please see above.

---

> ### Author Response · Authors · 2025-11-21
> **Rebuttal by Authors**
>
> > Q1: The authors add the regularization on the Jacobian to meet the sparsity assumption. This may be inefficient for high dimensional data.
>
> A: Following prior work on identifying the latent representations [1-12], our goal is to learn a *low-dimensional* set of latent variables that generates high-dimensional observations.
> Accordingly, we deliberately focus on moderate latent dimensionalities rather than very high-dimensional latent vectors.
> The Jacobian regularizer is applied to the estimates of mixing function with respect to these latent variables, so its computational cost scales with the latent dimension.
>
> In light of your suggestion, we additionally evaluated the scalability of our method to higher latent dimensions on the synthetic dataset by varying the latent dimensionality $N \in \{8, 12, 18\}$, while keeping the network architecture, training protocol, and all other hyperparameters fixed. Table A reports the MCC and compares with IndVAE [1]:
>
> Table A: The MCC score with different $N$
> | $N$  | IndVAE          | Ours           |
> |------|-----------------|----------------|
> | $8$  | $0.64 \pm 0.06$ | $0.80 \pm 0.02$ |
> | $12$ | $0.51 \pm 0.03$ | $0.68 \pm 0.05$ |
> | $18$ | $0.47 \pm 0.04$ | $0.61 \pm 0.05$ |
>
> Even as the latent dimension increases, our method consistently achieves higher MCC than IndVAE, indicating that the Jacobian-based sparsity regularization remains effective and that our approach scales well to  higher-dimensional latent spaces.
>
> > Q2: There are three regularization terms, hence three hyperparameters for their weights. There is no sensitivity analysis of them.
>
> A: In our implementation on the synthetic dataset for the results in Table 1 of the paper, we set the weight of the sparsity regularizer $\lambda$ to $1$ and the KL weights to $\beta_1=\beta_2=0.02$. Following your suggestion, we conducted a sensitivity analysis in which we vary one of these three hyperparameters at a time while keeping the others fixed at their default values. For all experiments we use the same network architecture, batch size, number of epochs, learning rate, and training protocol as in the main experiments. The numbers reported below are MCC scores on the synthetic dataset.
>
> Table B: The hyperparameter sensitivity analysis
> | Hyperparameter                    | Value          | Score (MCC)      |
> |-----------------------------------|----------------|------------------|
> | Sparsity weight $\lambda$        | $\lambda = 1$  | $0.87 \pm 0.04$  |
> | Sparsity weight $\lambda$        | $\lambda = 0.1$| $0.82 \pm 0.03$  |
> | Sparsity weight $\lambda$        | $\lambda = 0.01$| $0.70 \pm 0.07$ |
> | Sparsity weight $\lambda$        | $\lambda = 10$ | $0.68 \pm 0.05$  |
> | KL weight $\beta_1$              | $\beta_1 = 0.02$| $0.87 \pm 0.04$ |
> | KL weight $\beta_1$              | $\beta_1 = 1$   | $0.73 \pm 0.02$ |
> | KL weight $\beta_1$              | $\beta_1 = 0.001$| $0.65 \pm 0.04$|
> | KL weight $\beta_2$              | $\beta_2 = 0.02$| $0.87 \pm 0.04$ |
> | KL weight $\beta_2$              | $\beta_2 = 1$   | $0.82 \pm 0.01$ |
> | KL weight $\beta_2$              | $\beta_2 = 0.001$| $0.78 \pm 0.06$|
>
> Table B shows the default setting $(\lambda, \beta_1, \beta_2) = (1, 0.02, 0.02)$ obtains the best results. In light of your suggestion, we incorporate the sparsity weight $\lambda$ in Eq.12 in the paper revision.

---

> ### Author Response · Authors · 2025-11-21
> **Rebuttal by Authors**
>
> > Q3: For real data analysis, there is only one data set. Furthermore, it consists of images. I am afraid the MLP architecture may not be optimal for images.
>
> A: Our work's primary contribution is theoretical - establishing identifiability guarantees for the data generating process with dependent noise under generalized dependency structure. While additional real-world experiments could provide further validation, evaluating identifiability on real-world data is inherently challenging due to the absence of ground-truth latent variables.
>
> In light of your suggestion, we expand our real-world evaluation beyond the original dataset and consider two additional person index classification benchmarks. Specifically, we use the SYSU-MM01 dataset [13] and the RobotPKU dataset [14]. We only use the RGB modality, since our focus is not on cross-modal person re-identification. SYSU-MM01 contains RGB images of $491$ identities from $6$ cameras, with a total of $30{,}071$ images. RobotPKU contains more than $16{,}000$ RGB images of $180$ identities, captured under dynamic robotic viewpoints. These datasets thus provide diverse and challenging real-world testbeds for our framework.
>
> For performance comparison, we follow the same person index classification protocol as in our main experiment and compare against several state-of-the-art methods, including those has been compared in the paper, like GTL, AGW, TransReID, CLIPReID, MCRL and IndVAE. Also, we compare with the leading studies on RobotPKU benchmark, such as LDP-net [13], Style [14]. Tables C and D report the Top-1 classification accuracy.
>
> Table C: Comparison of Top-1 Accuracy on the RobotPKU dataset.
> | Methods   | Acc                         |
> |---------- |-----------------------------|
> | AGW       | $87.6 \pm 0.8$              |
> | TransReID | $90.2 \pm 0.9$              |
> | CLIPReID  | $91.7 \pm 1.1$              |
> | GTL       | $93.9 \pm 0.5$              |
> | MCRL      | $94.5 \pm 1.0$              |
> | IndVAE    | $95.8 \pm 0.8& |
> | Ours      | $97.0 \pm 0.5$          |
>
>
> Table D: Comparison of Top-1 Accuracy on the SYSU-MM01 dataset
> | Methods  | Acc                         |
> |----------|-----------------------------|
> | LDP-net  | $91.7 \pm 1.1$              |
> | Style    | $92.8 \pm 0.8$              |
> | CLIPReID | $94.1 \pm 1.0$              |
> | GTL      | $95.7 \pm 0.4$              |
> | MCRL     | $96.4 \pm 0.8$              |
> | IndVAE   | $96.8 \pm 0.5$  |
> | Ours     | $97.6 \pm 0.5$          |
>
> Following prior work on causal representation learning from images [2-10], we do **NOT** apply an MLP directly to raw pixels.
> Instead, we use a pretrained CLIP visual encoder with a ResNet-50 backbone, which we top a full-connected layer over the visual encoder and fine-tune to extract $1280$-dimensional feature vectors that serve as the observed variables $x$ in our data-generating process. Our MLP-based implementations operate on these high-level visual features, which is standard practice and computationally efficient. To further examine the effect of the architecture choice, we replace the MLP in our framework on the MSTM-17 dataset with a single-layer Gated Recurrent Unit (GRU) [15] using the same hidden dimension (the classifier architecture and all training and evaluation protocols remain unchanged, and we set $N=M=32$ for fair comparison). The resulting Top-1 accuracies are: GRU: $94.9 \pm 0.4$ versus our original MLP-based model: $94.4 \pm 0.7$. The GRU improves the classification results slightly against the MLP architecture. Overall, our method consistently outperforms strong baselines across three real-world person identity datasets.
>
> In light of your suggestions, we have incorporated all the additional results and discussions in Sec.D.3 in the appendix.

---

> > ### Author Response · Authors · 2025-11-21
> > **Rebuttal by Authors**
> >
> > --*Reference:*
> >
> > 1 Hu. Instrumental variable treatment of nonclassical measurement error models. Econometrica, 2008.
> >
> > 2 Kong, et al. Partial disentanglement for domain adaptation. ICML 2022
> >
> > 3 Sun,et al. Causal representation learning from multimodal biomedical observations. ICLR 2025
> >
> > 4 Khemakhem, et al. Variational autoencoders and nonlinear ica: A unifying framework. AISTAS, 2020
> >
> > 5 Chen, et al. Caring: Learning temporal causal representation under non-invertible generation process. ICML 2024
> >
> > 6 Lachapelle, et al. Synergies between disentanglement and sparsity: Generalization and identifiability in multi-task learning. ICML, 2023
> >
> > 7 Zheng, et al. Generalizing nonlinear ica beyond structural sparsity. NeurIPS 2023
> >
> > 8 Li, et al. Learning causal domain-invariant temporal dynamics for few-shot action recognition. ICML 2024
> >
> > 9 Li, et al. Identification of intermittent temporal latent process. ICLR 2025
> >
> > 10 Ng, et al. A general representation-based approach to multi-source domain adaptation. ICML 2025
> >
> > 11 Wu, et al. Rgb-infrared cross-modality person re-identification. CVPR 2017
> >
> > 12 Liu, et al. Online rgb-d person re-identification based on metric model update. CAAI Transactions on Intelligence Technology 2017
> >
> > 13 Zhou, et al. Revisiting prototypical network for cross domain few-shot learning. CVPR 2023
> >
> > 14 Fu, et al. Styleadv: Meta style adversarial training for cross-domain few-shot learning. CVPR 2023
> >
> > 15 Goodfellow, et al. Deep learning, volume 1. MIT Press 2016

---

### Official Review · Reviewer_5L1z · 2025-11-04

**Soundness:** 2
**Presentation:** 3
**Contribution:** 2
**Rating:** 4
**Confidence:** 4

**Summary:**

The paper studies identifiability of latent variables when (i) observations remain dependent given latents and (ii) the noise depends on the latents. It offers a two‑step theory: subspace identifiability from several measurement and component‑wise identifiability from structural sprsity/sufficient changes. An unsupervised VAE-style model is proposed to model the process. Experiments on synthetic dataset and real-world dataset on MSMT17 supports the claims.

**Strengths:**

1. Most prior identifiability results assume conditional independence, yet this paper allows dependence, which is significant.
2. Theory and identifibility results are clear and easy to follow.
3. The model matches the theory well.

**Weaknesses:**

1. The assumptions need some illustration to strengthen the readability. Is it possible to provide some illstrative explaination about the assumptions about under what circumstance then can be assured?
2. The baseline results reported in the table 1 are higher than those in the original papers. Is there any difference on the setting or it is Top-5 accuracy instead?
3. The theory is not new, i.e., Theorem 1 is mentioned in [1],  Theorem 2 is mentioned in [2], and Theorem 3 in [3]. The difference and similarity should be mentioned in the paper.

[1] Identification of Nonparametric Dynamic Causal Structure and Latent Process in Climate System

[2] On the identifiability of nonlinear ica: Sparsity and beyond.

[3] Partial Identifiability for Domain Adaptation.

**Questions:**

1. The identifiability is proven on $z$ only. Does $\epsilon$ identifiabile as well?
2. Theorem 2 and theorem 3 seem to be parallel. Each of them can be used independently to achieve component-wise identifiability. Can them be combined together to achieve stronger identifiability results?

---

> ### Author Response · Authors · 2025-11-21
> **Rebuttal by Authors**
>
> > Q1: The assumptions need some illustration to strengthen the readability. Is it possible to provide some illstrative explaination about the assumptions about under what circumstance then can be assured?
>
> A: We expanded the discussion of the main assumptions as follows.
>
> 1. Assumption $i$: generalized dependency structure
>
> This assumption is motivated by practical scenarios such as sensor networks or medical imaging.
> A concrete example is the chest X-ray diagnosis scenario described in the introduction from L.36 to L.42 in the paper revision: the goal is to infer a patient’s latent lung-cancer state $z$ from pixel intensities $x$. The patient’s inspiratory level, which may itself depend on disease status, acts as an unobserved noise $\epsilon$ that perturbs multiple anatomical regions. If we partition the image into three regions $x_a,x_b,x_c$, these regions still remain correlated even after conditioning on $z$.
> Hence, assumption (i) is satisfied.
>
> 2. Assumption $ii$: injective conditional operators $L_{x_a|z}$, $L_{z|x_c}$, $L_{x_a|x_c}$.
>
> This is a *mild* functional completeness condition in nonparametric factor analysis like [1,2]. Intuitively, it guarantees that the conditional pdf of $p(x_a|z),p(x_b|z),p(x_c|z)$ varies as $z$ changes. These properties hold generically when $p(x_a|z),p(z|x_c), p(x_a|x_c)$ are smooth and non-degenerate.
>
> 3. Assumption $iii$: non-degenerate spectrum of $L_{x_a,x_b|x_c}L_{x_a|x_c}^{-1}$
>
> Intuitively, each $z$ induces a unique eigenvalue in $L_{x_a,x_b|x_c}L_{x_a|x_c}^{-1}$, so the eigenvalue degeneracy can be avoided. Such spectral non-degeneracy is a generic condition for injectivity operators, which are assumed in Assumption $ii$.
>
> Assumption $iii$ also requires the eigenvalues of $L_{x_a,x_b|x_c}L_{x_a|x_c}^{-1}$ to have the cardinality equal to that of $(L_{x_b | z}$. This assumption holds when the dimension of $p(x_b|z)$ equals with the eigenvalue of $L_{x_a,x_b|x_c}L_{x_a|x_c}^{-1}$. We explain the circumstance that this assumption holds by an example. In our synthetic construction in Eq.49 in the paper, there are exactly three independent latent sources $z_1,z_2,z_3$, and $x_b$ depends smoothly and non-trivially on all of them through $R$ and $\sigma$. Hence the family $\{p(x_b|z)\}$ is genuinely 3–dimensional, and $L_{x_b|z}$ has three non-zero eigenvalues. The only difference between $x_a$ and $x_b$ also comes from the three-dimensional $z$. Consequently $L_{x_a,x_b|x_c}L_{x_a|x_c}^{-1}$ has exactly the a three-dimensional non-zero eigenvalues as well. Since $R$ is full rank with small nonzero off-diagonal entries, our $R$ enables these three eigenvalues are all distinct and well separated. Thus Assumption $iii$ is satisfied by construction.
>
>
> 4. Assumption $iv$: self-adjointness of $L_{x_a|z} L_{x_b|z} L_{x_a|z}^{-1}$
>
> Assumption $iv$ is mild and is satisfied when:
> 1. $L_{x_a|z}$ is injective by Assumption $ii$, so it defines an isomorphism between the Hilbert space $\mathcal H_z$ and its range $\mathrm{ran}(L_{x_a|z}) \subset \mathcal H_{x_a}$;
> 2. $p(x_b|z)$ is symmetric.
>
> Let us explain with with the example of our data synthesizing process in Eq. 49 in Sec.C.1 in paper revision. We enforce this  assumption directly through the mixing matrix $R$. We first fix an invertible row block $R_a\in\mathbb R^{3\times 3}$ corresponding to $x_a$, then choose any symmetric matrix $B\in\mathbb R^{3\times 3}$ with distinct real eigenvalues, such as $B=\mathrm{diag}(1,2,3)$, and set $R_b:= B R_a$, for the row associated with the $x_b$. On the space of $z$, the transform $L_{x_a|z}L_{x_b|z}L_{x_a|z}^{-1}$ is then represented by the matrix $B = R_b R_a^{-1}$, which is symmetric by construction and hence defines a self-adjoint operator.
>
> 5. Assumption $v$: eigenvalue gap and bounded perturbation
>
> Assumption $v$ holds when $L_{x_a,x_b|x_c}L_{x_a|x_c}^{-1}$ has distinct eigenvalue, which is guaranteed by Assumption $iii$, and the eigengap between $L_{x_a,x_b|x_c}L_{x_a|x_c}^{-1}$ and $L_{x_a|z} L_{x_b|z} L_{x_a|z}^{-1}$ is small.
> We provide an example to explain when this assumption holds. In the synthetic experiment we keep $R$ in Eq.49 with only a few off-diagonal and nonzero entries. Since all the operators involved depend on the entries of $R$, the upper bound of the perturbatoin operator $\overline{Per}$ remains strictly smaller than the eigen-gap for $R$ in a sufficiently small neighbourhood of the diagonal choice. Therefore, Assumption $v$ holds.
>
> 6. Assumption $vi$: existence of an operator $M$ such that $M(L_{x_b|z}) = M(L_{x_b|\hat h(z)}) = t(z)$
>
> Assumption $vi$ restricts the transformation $t \in T : \mathcal{Z} \to \mathcal{Z}$ to be smooth and invertible, a standard constraint in nonlinear identifiability results  [3-13]. It holds whenever the map $z \mapsto L_{x_b|z}$ is injective and smooth, which is a generic situation when the conditional distributions $p(x_b|z)$ vary smoothly and non-degenerately with $z$.

---

> ### Author Response · Authors · 2025-11-21
> **Rebuttal by Authors**
>
> > Q2: The baseline results reported in the table 1 are higher than those in the original papers. Is there any difference on the setting or it is Top-5 accuracy instead?
>
> A: The higher reported numbers is due to: 1. a different evaluation *metric*; and 2. a different experimental *setup*. We have stressed that all numbers in Table 1 are *Top-1 classification accuracies*, not Top-5 in the caption of Table 1.
>
> - **Different Metric:**
>
> As stated in L.430–431 of the paper, in our real-world experiment we cast the problem as a person-index *classification* task. We first train a VAE with our objective in Eq.~(12) to obtain a latent representation $z$, and then train a classifier on $z$ to predict the identity label $y$, i.e., to model $p(y |z)$. We then report the *Top-1 classification accuracy*.
>
> By contrast, AGW, TransReID and CLIP-ReID are originally evaluated on MSMT17 as *person re-identification* methods, using *Rank-1*: given a query, one computes feature similarities to a gallery and measures whether the correct identity is ranked first.
> Top-1 classification accuracy and Rank-1 ReID accuracy are therefore not directly comparable.
> In our experiments, we use the public code of these methods to extract features on MSMT17 and then train an identity classifier on top of these features, reporting Top-1 classification accuracy under our protocol.
>
> - **Different Setup:**
>
> We have stated our experiment setup and dataset from L.430 to L. 456 in the original paper . We summarize in the following:
> For fairness, all methods in Table 1 in the paper are trained under the *same* classification protocol on MSMT17: we randomly split the data into 60\% for training, 20\% for validation and 20\% for test, with the same set of identities present in both splits. We train each backbone using its official implementation (on the training split) to obtain features, and then append a classifier to predict person identities. In contrast, the standard ReID protocol uses disjoint identities between train and test and evaluates retrieval over a gallery, and [14] utilizes a few-shot classification setting. This change from the setup of ReID / few-shot classification to closed-set classification naturally yields higher absolute accuracy values.
>
> - **Rationale for Using Classification:**
>
> Classification has been widely used to empirically assess identifiability, e.g., [6,15-21]. Following this line of work, we use a closed-set classification task to test the identifiability guarantees: after learning $z$, we
> directly measure how well $z$ predicts the identity label $y$.
> High Top-1 classification accuracy thus provides a clean and direct proxy for how much identity information is preserved in $z$.
>
> In contrast, standard ReID benchmarks treat test identities as “unseen” and evaluate pipeline components such as metric learning, similarity computation, and nearest-neighbor search, which introduce additional approximation and estimation errors that are orthogonal to our goal. To isolate and evaluate identifiability of the latent variables, we therefore deliberately adopt the simpler and more controlled classification setup rather than the conventional ReID protocol.

---

> ### Author Response · Authors · 2025-11-21
> **Rebuttal by Authors**
>
> > Q3: The theory is not new, i.e., Theorem 1 is mentioned in [22], Theorem 2 is mentioned in [23], and Theorem 3 in [4]. The difference and similarity should be mentioned in the paper.
>
> A: We would like to clarify that our key contribution -- Theorem 1 -- has **NOT** been presented in [22].
> In the following, we first analyze the difference between [22] and our Theorem 1, then we recapitulate the reason [23,4] cannot handle our generalized dependency setup.
>
> 1. The difference between our Theorem 1 and [22]
>
> *A. Modeling setup:*
>
> Our work explicitly allows any arbitrary generalized dependency structure among the observed variables conditional on $z$, formalized in Assumption $i$ of Theorem 1 as $p(x | z) \neq p(x_a | z) p(x_b | z) p(x_c | z)$. In contrast, Eq. A2 in [1] explicitly imposes that ``$x_{t-1}, x_t, x_{t+1}$ are conditional independent given $z_t$'', **which directly contradicts the generalized dependency structure defined in our work**.
>
> *B. Identifiability result:*
> Under this more general dependence, our Theorem 1 establishes subspace identifiability via the operator equation (Eq. (4) in our paper):
>
>  $L_{x_a |z} L_{x_b |z} L_{x_a |z}^{-1}
>       = L_{x_a,x_b |x_c} L_{x_a |x_c}^{-1} + \mathrm{Per}$,
>
> where $\mathrm{Per} \neq 0$ captures the deviation induced by the generalized dependency. Handling this nonzero perturbation term requires a careful perturbation-theoretic analysis to show that the relevant eigenspaces remain identifiable.
> In [22], the bases of their Theorem 1 (Eq. A11) relies on exact conditional independence so that no perturbation term appears. Our theorem therefore strictly generalizes this situation and provides an identifiability guarantee in the presence of arbitrary dependence among $x_a,x_b,x_c$.
>
> In summary, Our Thm.1 provides a new and more robust subspace identifiability result for $z$ under a generalized dependent-noise structure that violates the conditional-independence assumptions of [22].
>
> In light of your suggestion, we have added [22] to our references and disucss the difference in our introductions from L.34 to L.35, as well as the description from L.54 to L.56.
> **We also welcome further discussion if the reviewer can specify the exact conclusion from our Theorem 1 have been explicitly introduced in [22]**.
>
> 2. The relation between our Theorems 2,3 and [23,4].
>
> We have discussed the connections and differences between Theorem2 and [23] in L.241–L.244 in our original paper (L.238 to L.243 in paper revision), and between Theorem3 and [4] in L.279–L.284 in original paper (L.279–L.284 in paper revision), respectively. In the following, we re-summarize and elaborate the differences further:
>
> There exists an important difference in terms of data generating process between our work and [23,4]:
> [23,4] does not include the dependent noise.
> Our analysis proceeds in two stages: (1) Theorem 1 establishes subspace identifiability of $z$ under our generalized dependent-noise model $x = g(z,\epsilon)$ with $\epsilon = e(z,\eta)$.
> This disentangles $z$ from the noise in the sense that there exists an invertible map $h$ with $\hat{z} = h(z)$ even when the noise depends on $z$. (2) Given this subspace identifiability, Theorems 2 and 3 then arrives at the result to component-wise identifiability, in the spirit of the component results, but *under the more general dependent-noise assumptions ensured by Theorem1*.
>
> By contrast, [23] and [4] work in a strictly more restrictive noise-free setting, which corresponds to $x=g(z)$ in our notation and does not allow $\epsilon = e(z,\eta)$ that we consider. Hence, the component-wise identifiability results in [23,4] cannot be directly applied in our setting without first establishing Theorem1.
>
> In light of your suggestions, and given the fact that our key contribution lies in Theorem 1, we have renamed Theorem 2 and 3 by "Corollary 1'' and "Corollary 2'', respectively, to avoid any confusion.

---

> > ### Author Response · Authors · 2025-11-21
> > **Rebuttal by Authors**
> >
> > > Q4: The identifiability is proven on $z$ only. Does $\epsilon$ identifiabile as well?
> >
> > A: A simple thought would be, if the analysis from [1] can be applied for $\epsilon$, and we also impose Assumptions $i$–$vi$ of Thm.~1 with $z$ replaced by $\epsilon$, and additionally choose the partition $x=(x_a,x_b,x_c)$ in Assumption $i$ so that the corresponding conditional operators are injective for both $p(x|z)$ and $p(x|\epsilon)$, then the same proving strategy can yield identifiability of $\epsilon$ up to an invertible transformation $h_{\epsilon}$, i.e., $\hat{\epsilon}=h_{\epsilon}(\epsilon)$.
> >
> > Moreover, combined with Thm.~1, the existence of $h_{\epsilon}$ implies block-wise identifiability of $(z,\epsilon)$, namely $\hat{z}=h(z)$ and $\hat{\epsilon}=h_\epsilon(\epsilon)$, if the following assumption holds:
> > *There exists an smooth, invertible transformation $f$ between $z,\epsilon$ and $\hat{z},\hat{\epsilon}$. Also, the diagonal entries of the jacobian of $f$ coincides with $\frac{\partial \hat{z}}{\partial z} $ and $ \frac{\partial \hat{\epsilon}}{\partial \epsilon}$, respectively.*
> >
> > In light of your suggestions, we have added a Corollary in Sec.B.5 in the Appendix to clarify this point.
> >
> > > Q5: Theorem 2 and theorem 3 seem to be parallel. Each of them can be used independently to achieve component-wise identifiability. Can them be combined together to achieve stronger identifiability results?
> >
> > A: We realize there exists a work [24] that studies combining the assumptions underlying our Theorem2 and Theorem3 (in their notation) in order to obtain component-wise identifiability of $z$ in a *noise-free* model.
> > Please refer to [24] for the details.
> >
> > Our focus, however, is fundamentally different. We consider a data-generating process with *dependent noise* under the generalized dependency structure and establish identifiability results in this more general model.  [24] assumes noise-free observations and does not model dependent noise, thus cannot handle the generalized dependency structure that our theorems are designed to address.
> >
> >
> > *--Refence:*
> >
> > 1 Hu. Instrumental variable treatment of nonclassical measurement error models. Econometrica, 2008.
> >
> > 2 Mattner. Some incomplete but boundedly complete location families. The Annals of Statistics, 1993.
> >
> > 3 Liang, et al. Causal component analysis. NeurIPS 2023
> >
> > 4 Kong, et al. Partial disentanglement for domain adaptation. ICML 2022
> >
> > 5 Sun,et al. Causal representation learning from multimodal biomedical observations. ICLR 2025
> >
> > 6 Kong, et al. Identification of nonlinear latent hierarchical models. NeurIPS 2023
> >
> > 7 Zheng, et al. Nonparametric factor analysis1 and beyond. AISTATS 2025
> >
> > 8 Morioka, et al. Connectivity-contrastive learning: Combining causal discovery and representation learning for multimodal data. AISTATS 2023
> >
> > 9 Fu, et al. Identification of nonparametric dynamic causal model and latent process for climate analysis. Arxiv, 2025
> >
> > 10 Khemakhem, et al. Variational autoencoders and nonlinear ica: A unifying framework. AISTAS, 2020
> >
> > 11 Morioka et al. Causal representation learning made identifiable by grouping of observational variables. ICML 2024
> >
> > 12 Hyvarinen, et al. Unsupervised feature extraction by time-contrastive learning and nonlinear ica. NeurIPS 2016
> >
> > 13 Reizinger, et al. Jacobian-based causal discovery with nonlinear ICA. TMLR 2023
> >
> > 14 Yang, et al. Crossmodal few-shot learning: a generative transfer learning framework. Arixv 2025
> >
> > 15 Li, et al. Subspace identification for multi-source domain adaptation. NeurIPS 2023.
> >
> > 16 Chen, et al. Caring: Learning temporal causal representation under non-invertible generation process. ICML 2024
> >
> > 17 Lachapelle, et al. Synergies between disentanglement and sparsity: Generalization and identifiability in multi-task learning. ICML, 2023
> >
> > 18 Zheng, et al. Generalizing nonlinear ica beyond structural sparsity. NeurIPS 2023
> >
> > 19 Li, et al. Learning causal domain-invariant temporal dynamics for few-shot action recognition. ICML 2024
> >
> > 20 Li, et al. Identification of intermittent temporal latent process. ICLR 2025
> >
> > 21 Ng, et al. A general representation-based approach to multi-source domain adaptation. ICML 2025
> >
> > 22 Fu, et al. Identification of Nonparametric Dynamic Causal Structure and Latent Process in Climate System. Arxiv 2025
> >
> > 23 Zheng, et al. On the identifiability of nonlinear ica: Sparsity and beyond. NeurIPS, 2022
> >
> > 24 Li, et al. Synergy between sufficient changes and sparse mixing procedure for disentangled representation learning. ICLR 2025

---

### Author Response · Authors · 2025-11-29
**Author Final Remarks**

Dear AC and SAC,

We are providing a brief and self-contained summary of our contributions and our rebuttals.

**Contributions:**

- **Theoretically**: We establish the identifiability result that recovers the latent variables $z$ even when the observables exhibit arbitrary conditional dependence given $z$. In contrast to prior work, our setup does *NOT* assume conditional independence of the observed variables given the latents.

We are encouraged by the comments that our theoretical results "is significant" (5L1z), "renders more flexibility"(Mn5G), "advancement" with "novel assumptions" (Dt9i).

- **Practically**: We develop a variational-inference–based estimator that learns both the data-generating process and the latent variable $z$, yielding consistent improvement across both synthetic and real-world experiments.

The reviewers consider our results "show the effectiveness of these improvement" (Mn5G), "promising" (Dt9i) and "thorough" (TTFZ).

**Rebuttal Summary:**

- **Novelty w.r.t. [1,2]** (Reviewer 5L1z, Reviewer Dt9i): Both [1,2] impose conditional independence of observables given the latents. Thus, the comments from Reviewer 5L1z and Reviewer Dt9i that our conditional-dependence setup and Theorem 1 containing in [1,2] are *factually incorrect*.

- **Real-world experimental procedure** (Reviewer 5L1z, Reviewer Dt9i):
*We explained our detailed real-world experimental setup in Sec.7.2 in our paper, including (1) the task definition, (2) the evaluation protocol, and (3) the metrics*. We reiterate them in rebuttal and clarify how these differ from the original settings of the comparison methods. Both reviewers obviously misunderstand our paper.

- **Difference between our work and independent-noise setups** (Reviewer TTFZ): We clarified that our work does not impose the structural assumptions required in the standard independent-noise model, and our setup is more challenging for identification problem. Notably, given our work focuses on identifying $z$ under arbitrary conditional dependence of observations, such "primary concern" from Reviewer TTFZ *deviates from our scope*.

- **Discussions of Assumptions** (Reviewer 5L1z, Reviewer TTFZ):
We expanded the discussion of our main assumptions, providing intuitive explanations and concrete scenarios where they hold. We also explained how each assumption is satisfied *by construction* in our synthetic data-generating process.

- **Corollary 2,3** (Reviewer 5L1z, Reviewer TTFZ):
our Corollary 2,3 are built upon our Theorem 1, which [3,4] can *NOT* be directly applied given they only assume a noise-free setup.

- **Additional experiments** (Reviewer Mn5G):
We conducted additional analyses, including ablation studies, hyperparameter sensitivity experiments, and experiments on additional real-world datasets.

We kindly request that ACs and SACs take our rebuttal, additional results, and final remarks into account when making the decision on our submission, especially given that several critical concerns were based on *misunderstandings and misreadings* that we have directly addressed in rebuttal in details. We thus believe that, the ratings **should be adjusted accordingly**.

-- *References*

[1] Fu, et al. Identification of Nonparametric Dynamic Causal Structure and Latent Process in Climate System. Arxiv 2025

[2] Li, et al. Online time series forecasting with theoretical guarantees. NeurIPS 2025

[3] Kong, et al. Partial disentanglement for domain adaptation. ICML 2022

[4] Zheng, et al. On the identifiability of nonlinear ica: Sparsity and beyond. NeurIPS, 2022

---

### Meta-Review · Area_Chair_q2vA · 2026-01-07

**Summary:**

This paper removes the standard conditional-independence assumption by proving identifiability of latent representations even when observations exhibit arbitrary dependencies given the latents.

It introduces a two-step framework, subspace identifiability via perturbation-based factor analysis and component-wise identifiability under sparsity or latent variability, yielding an unsupervised method with theoretical guarantees validated empirically.

**Reviewer Concerns:**

- novelty: whether the main theorems were already in prior papers and proper discussion of closely related contemporaneous work.

- baseline numbers different from reported in original paper (clarified)

- the main theorem’s setup needs more motivation, eg how to partition observations and whether assumptions are satisfied in the synthetic setup

- dependent noise aspect might be reformulated away into an independent-noise view

- unclear notation and logical steps in Theorem 1 (resolved)

- Jacobian sparsity regularization cost at higher dimensions, lack of hyperparameter sensitivity analysis, and the real-data evaluation (resolved)

**Reviewer Scores:**

Reviewer ttfz were able to actively participate in the discussion but was not convinced of the core problem formulation.

Other reviewers might marginally adjust their opinion.

---

### Decision · Program_Chairs · 2026-01-26

Reject